# Dietary thiols accelerate aging of *C. elegans*

Ivan Gusarov[1], Ilya Shamovsky[1], Bibhusita Pani[1], Laurent Gautier[1], Svetlana Eremina[2], Olga Katkova-Zhukotskaya[2], Alexander Mironov[2], Alexander A. Makarov[2] & Evgeny Nudler [1,3✉]

Glutathione (GSH) is the most abundant cellular antioxidant. As reactive oxygen species (ROS) are widely believed to promote aging and age-related diseases, and antioxidants can neutralize ROS, it follows that GSH and its precursor, N-acetyl cysteine (NAC), are among the most popular dietary supplements. However, the long- term effects of GSH or NAC on healthy animals have not been thoroughly investigated. We employed *C. elegans* to demonstrate that chronic administration of GSH or NAC to young or aged animals perturbs global gene expression, inhibits *skn-1*-mediated transcription, and accelerates aging. In contrast, limiting the consumption of dietary thiols, including those naturally derived from the microbiota, extended lifespan. Pharmacological GSH restriction activates the unfolded protein response and increases proteotoxic stress resistance in worms and human cells. It is thus advantageous for healthy individuals to avoid excessive dietary antioxidants and, instead, rely on intrinsic GSH biosynthesis, which is fine-tuned to match the cellular redox status and to promote homeostatic ROS signaling.

[1] Department of Biochemistry and Molecular Pharmacology, New York University School of Medicine, New York, NY, USA. [2] Engelhardt Institute of Molecular Biology, Russian Academy of Science, Moscow, Russia. [3] Howard Hughes Medical Institute, New York University School of Medicine, New York, NY, USA. ✉email: evgeny.nudler@nyumc.org

-acetylcysteine (NAC) is a common thiol antioxidant that is converted to L-cysteine (Cys) and GSH in vivo, respectively, for protein synthesis and cellular redox maintenance. As NAC directly scavenges ROS, it can suppress various pro-inflammatory pathways[1]. Animal studies of ROS-induced pathologies suggest that NAC supplementation may attenuate atherosclerosis, reduce lung inflammation, pulmonary fibrosis, transplant rejection, and be useful for treating arthritis[1]. Currently, however, the only approved clinical applications of NAC have been to treat acute GSH depletion associated with acetaminophen overdose and to thin mucus[2,3].

Due to its apparent antioxidant and anti-inflammatory properties, NAC has acquired a celebrity status among food supplements. However, pathology models fail to justify NAC supplementation in healthy individuals. For example, NAC consumption nullifies the beneficial effects of exercise in rodents and humans[4]. One argument for NAC supplementation, which reprises Harman's free radical theory of aging[5], is that aging results from accumulation of ROS-mediated damage. The fundamental caveat to this theory is that the amount of ROS-mediated damage required to shorten lifespan is usually never achieved without exposure to toxins or radiation[6]. It has become apparent that although high levels of oxidants can, indeed, be detrimental to living organisms, a low, physiological level of ROS is necessary for cellular signaling and delays aging[6–8]. As much as dietary antioxidants can alleviate acute stress, their chronic supplementation to healthy animals may interfere with normal ROS signaling, compromise homeostasis, and eventually accelerate, rather than decelerate, aging[4].

Here we report an investigation into the effects of prolonged supplementation and chronic restriction of the major cellular thiol antioxidants, NAC and GSH, on C. elegans physiology and lifespan. Nematodes proved to be a powerful model to study metabolic and signaling pathways that control aging. The high evolutionary conservation of these pathways, including the unfolded protein response (UPR) which we found to be affected by dietary thiols in worms and human cells, suggests that the results presented here can be extrapolated to higher organisms as well.

## Results

**Dietary thiols shorten the C. elegans lifespan.** To study the effect of dietary thiols on C. elegans aging we first transferred L4 stage worms to freshly prepared NGM plates spotted with E. coli OP50 and supplemented with pH-adjusted NAC. Surprisingly, under the specified conditions, not only did NAC fail to prolong the C. elegans lifespan, but it significantly and dose-dependently shortened the median animal lifespan (Fig. 1a). Such a negative effect of NAC on C. elegans aging was eliminated if we applied the chemical to E. coli OP50 seeded NGM plates two days prior to transferring the worms (Supplementary Fig. 1a), indicating that NAC was rapidly metabolized by bacteria and/or oxidized. NAC supplementation mildly extended the lifespan in the presence of 100 μM FUDR (Supplementary Fig. 1b), suggesting inadvertent drug interactions. Remarkably, supplementing C. elegans with their own natural thiol, GSH, also shortened their lifespan (Fig. 1b), demonstrating that exogenous thiol antioxidants can be detrimental to healthy animals.

To eliminate a possible indirect effect of exogenous thiols mediated by bacterial metabolism, we repeated the NAC experiment using dead bacteria. Irrespective of whether we killed E. coli by heat (Fig. 1c) or antibiotic treatment (Fig. 1e), addition of NAC shortened the C. elegans lifespan. As with live bacterial diet, a higher concentration of NAC further exacerbated its negative effect on the lifespan (Fig. 1c). These results indicate that

chronic administration of NAC to otherwise healthy animals has a concentration-dependent negative effect on their lifespan.

It has been generally assumed that older animals suffer from increasing oxidative stress and, therefore, should benefit from dietary antioxidants[9,10]. To test this conjecture, we treated worms with NAC at day 13 of adulthood, instead of at stage L4. Surprisingly, NAC also shortened the lifespan of these animals; albeit to a lesser extent as compared to the animals treated from stage L4 (Fig. 1d), suggesting that the pro-aging effect of NAC correlates with the duration of its exposure.

To verify that NAC-mediated shortening of the lifespan was thiol-specific, and not the result of an amino acid overconsumption, we examined the effect of N-acetyl serine (NAS). The same concentration of NAS had no effect on the animal lifespan (Fig. 1e), confirming that the detrimental effect of NAC was specific to its reduced thiol group.

**C. elegans are overdosed with GSH on a live E. coli diet.** It is well established that C. elegans live substantially longer if fed dead bacteria (DB) instead of live bacterial (LB)[11]. We noted that C. elegans fed DB in the presence of 15 mM NAC had a shortened lifespan comparable to those fed LB without NAC supplementation (Fig. 1a, c). Live E. coli is an enormous reservoir of reduced GSH. Its intracellular concentration of GSH is ~17 mM, 95% of which is in a reduced form[12]. C. elegans efficiently grind and lyse E. coli to release bacterial GSH into their intestine. Therefore, if reared on live E. coli, the nematodes must be constantly exposed to the high level of dietary GSH, which could limit their lifespan.

To test this hypothesis, we first examined the level of bacterial thiols in LB and DB. After 3 days of incubation on NGM plates the overall levels of reduced thiols and GSH became lower in DB (Fig. 2a), indicating that C. elegans fed LB are exposed to a higher level of reduced thiols. We next measured the level of reduced thiols in worm extracts and stained live C. elegans with thiol-specific fluorescent dye ThioFluor 623 (Fig. 2b, d, e). As expected, C. elegans fed DB had a lower concentration of total reduced thiols, lower GSH and lower in vivo thiol staining compared to animals fed LB or NAC-supplemented DB (Fig. 2b, d, e). These results demonstrate that endogenous thiols and GSH is not well controlled in C. elegans and may accumulate to a harmful level if provided in excess.

Similar to other oxidants[13], a specific thiol-oxidizing agent, diamide, has a contrasting effect on C. elegans aging: at low (5 mM) and high (15 mM) concentrations it, respectively, extends and shortens the lifespan of LB-fed animals[14], suggesting that both an excess and deficiency of cellular thiols are detrimental. Accordingly, 5 mM and 15 mM diamide, respectively, decreased the level of endogenous thiols mildly or almost completely (Fig. 2d, e). Worms reared on LB + 5 mM diamide or DB exhibit similar thiol levels (Fig. 2d, e) and an extended lifespan[11,14]. A short lifespan of worms reared on LB + 15 mM diamide[14] (Fig. 2e, d) indicates that further depletion of thiols, as compared to LB + 5 mM diamide or DB, is detrimental. Indeed, diamide shortened the lifespan of animals fed DB (Supplementary Fig. 1d, e).

In agreement with previously published data[15], NAC supplementation increased the resistance to oxidative stress (Supplementary Fig. 1c). Moreover, worms grown on LB or DB + NAC were more resistant to oxidative stress comparing to worms fed on DB (Fig. 2c), further supporting our conclusion that these animals accumulate excessive thiol antioxidants. Remarkably, although worms fed DB accumulated less reduced thiols and were more sensitive to paraquat (Fig. 2c), they lived longer (Fig. 1c). These results are consistent with our previous findings that a high

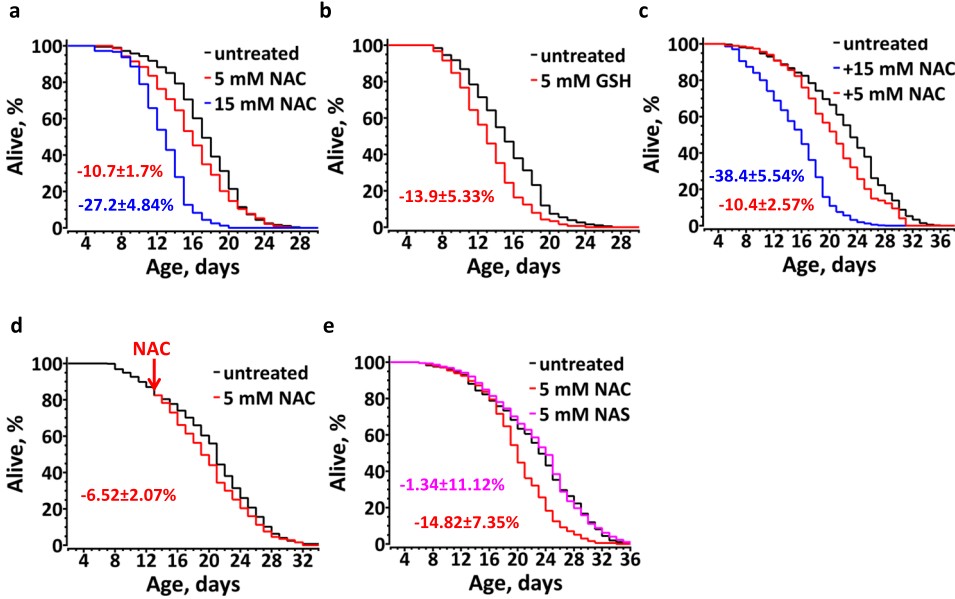

**Fig. 1 Thiol antioxidants shorten *C. elegans* lifespan.** Chronic supplementation of fresh NAC ($n = 234$(untreated), 159(5 mM NAC), 223(15 mM NAC)), (**a**) or GSH ($n = 175$(untreated), 168(5 mM GSH)), (**b**) limits the lifespan of live bacteria (LB)-fed *C. elegans*. **c** NAC dose-dependently shortens the lifespan of dead bacteria (DB)-fed *C. elegans* ($n = 525$(untreated), 263(5 mM NAC), 211(15 mM NAC)). **d** Supplementation of NAC to older animals mildly shortens their lifespan. N2 animals were fed DB until day 13 of adulthood; half of them were transferred to agar plates with 5 mM NAC ($n = 261$(untreated), 289(5 mM NAC)). **e** The SH-group of NAC accounts for accelerated aging. Worms fed DB treated with N-acetyl serine (NAS), which has an –OH, instead of a –SH ($n = 203$(untreated), 204(5 mM NAC), 221(5 mM NAS)). All graphs show the composite of three independent experiments (except **b**, which is a composite of two independent experiments). Average percentage change ±SD of the lifespan relative to an untreated control is indicated in color matching the corresponding curve. See also Supplementary Table 2.

sugar diet, while promoting GSH reduction and resistance to ROS, decreases *C. elegans* lifespan[14].

**Dietary thiols suppress anti-aging gene expression.** To elucidate the mechanism by which dietary thiols limit *C. elegans* lifespan, we studied the transcriptional response to NAC. First, we compared total transcriptomes of worms grown on DB + NAC and DB (Supplementary Data 1). According to gene ontology (GO) analysis, the genes determining adult lifespan, including well-characterized members of the DAF-16 regulon, *sod-3* and *mtl-1*, are among the most suppressed by NAC (Fig. 3a, b and Supplementary Data 1). Thus, the anti-aging insulin/IGF-signaling pathway mediated by DAF-16 is likely to be inhibited by NAC. Phenotype enrichment analysis also revealed that genes down-regulated by NAC are involved in oxidative and toxic metal stress resistance, suggesting that SKN-1, an ortholog of the mammalian master regulator of oxidative stress defense, NRF-2, is negatively regulated by NAC (Supplementary Fig. 2). The enrichment of dauer-related phenotypes further implicates DAF-16 (Supplementary Fig. 2). No aging-related GO categories were enriched among the genes upregulated by NAC (Supplementary Fig. 3). We therefore conclude that NAC shortens the lifespan by inhibiting anti-aging gene expression.

To substantiate this conclusion, we searched for DAF-16 and SKN-1 targets among the 1382 genes downregulated >2 fold by NAC in *C. elegans* fed DB (Supplementary Data 1). We found that 38% and 40% of genes, which are positively controlled, respectively, by DAF-16[16] and SKN-1[17] were repressed by NAC (Fig. 3c). The overlap between the genes downregulated by NAC and the SKN-1 (92 genes) or DAF-16 (189 genes) sets was statistically significant and over five-fold above chance (Fig. 3c). Integrative analysis of these datasets demonstrates that the most affected phenotypes were oxidative stress and cadmium

sensitivity (Supplementary Fig. 4a). It appears that NAC selectively suppresses a subset of DAF-16 and SKN-1-regulated genes involved in the defense against oxidative stress. Thus, paradoxically, dietary NAC, while increasing resistance against oxidative stress due to accumulation of antioxidant thiols, simultaneously blunts the endogenous cytoprotective gene network, resulting in a shortened lifespan.

As DAF-16 and SKN-1 themselves are under redox regulation[18,19], the dietary thiols could deactivate these transcription factors by reducing their redox sensitive partners. However, 5 mM NAC has an even stronger detrimental effect on *daf-16* animals (~20% lifespan decrease, Fig. 3d) as compared to wt (~10%, Fig. 1a), indicating that DAF-16 is required for the adaptation to a high thiol diet.

Our RNA-seq data demonstrate that the expression of GSH biosynthetic genes (Fig. 4a) and 14 *gst* genes are downregulated by thiol rich diets (Supplementary Data 1). A well-studied SKN-1-dependent gene, *gst-4*, was downregulated ~10-fold by NAC (Supplementary Data 1, Fig. 4a). Concordantly, the fluorescence of *gst-4::GFP* reporter animals was induced on DB diet (Fig. 4b, c). Moreover, dietary NAC and GSH decreased the *gst-4::GFP* expression on LB and DB (Fig. 4b, c and Supplementary Fig. 4b, c), implying that SKN-1 is inhibited by exogenous thiols. This interpretation is consistent with our finding that NAC failed to shorten the lifespan of *skn-1*-deficient worms (Fig. 4d). SKN-1 regulation can be bypassed by inactivating WDR-23, which normally drives SKN-1 to proteasomal degradation under non-stressed conditions[20]. Indeed, we found a significant overlap (RF = 1.5, *p* value = 2.07e−10) between the genes downregulated by NAC and upregulated in WDR-23-deficient worms (Supplementary Fig. 5a)[21]. Oxidative stress and cadmium sensitivity were among the most enriched phenotypes (Supplementary Fig. 5b). In agreement with these observations, NAC failed to shorten the lifespan of WDR-23-deficient animals (Supplementary Fig. 5c, d).

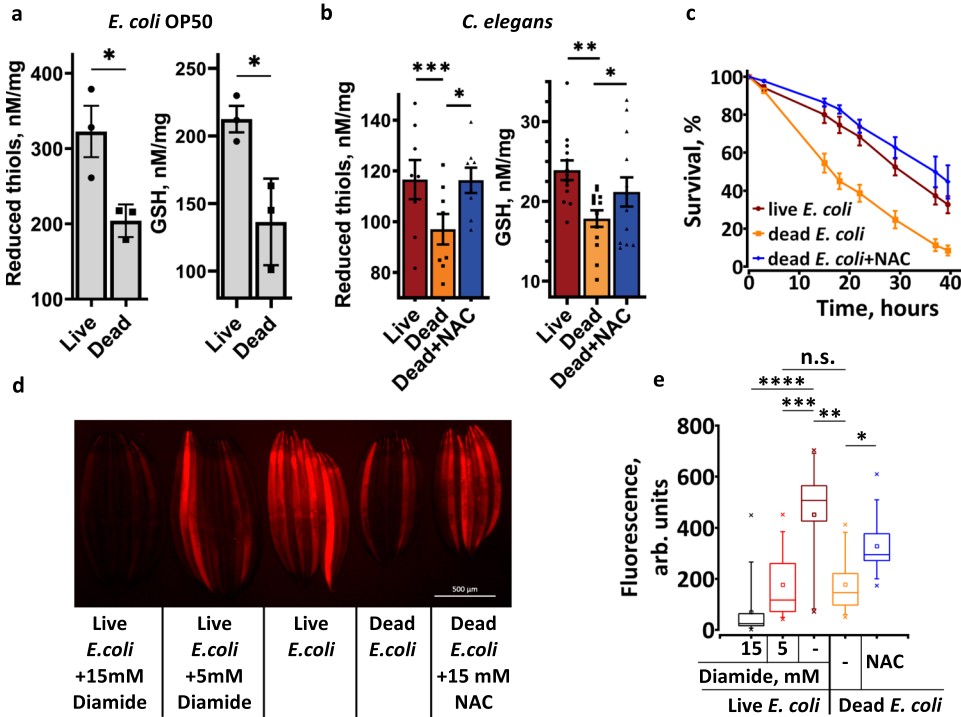

**Fig. 2 Thiol level in *C. elegans* depends on the diet. a** Killed bacteria have a lower level of reduced thiols. Live or dead (heat inactivated) *E. coli* OP50 spotted onto NGM plates. After three days, bacteria were scraped and the total amount of reduced thiols (left panel) and GSH (right panel) was determined. Error bars, mean ± SEM from three independent experiments. **b** The level of thiols in *C. elegans* depends on the thiols content in the diet. L4 stage animals were transferred to LB, DB or DB + NAC plates, incubated for three days, and the total amount of reduced thiols (left panel, *n*~800) and GSH (right panel, *n*~2600) was determined. Error bars, mean ± SEM from at least 8 independent experiments. See also Supplementary Table 3. **c** Thiols supplemented in the medium or acquired from live bacteria promote oxidative stress resistance. N2 (wt) animals were grown on live bacteria (LB), dead bacteria (DB), or DB + NAC plates. At day 2 of adulthood they were transferred to plates spotted with DB and supplemented with 150 mM paraquat. Error bars, mean ± SEM from three independent experiments (*n* = 120). **d, e** Representative image and quantification of thiols staining in live *C. elegans*. L4 stage animals were transferred to LB or DB plates supplemented with chemicals as indicated, incubated for three days, picked, washed, and stained with ThioFluor 623 to detect intracellular reduced thiols. Box plots indicate median (middle line), 25th, 75th percentile (box) and 5th and 95th percentile (whiskers) as well as maximum, minimum and mean (single points). *n* = 94–128 worms over three independent experiments. See also Supplementary Table 4. In all graphs *p* values are: n.s. not significant; *$p < 0.05$; **$p < 0.01$; ***$p < 0.001$; ****$p < 0.0001$; two-tailed *t*-tests.

Taken together, our results demonstrate that SKN-1 inhibition is one of the major reasons for accelerated aging caused by a NAC-supplemented diet.

To gain further insight into the mechanism by which DB extends the *C. elegans* lifespan, we searched the DB-fed *C. elegans* transcription profile against the publicly available gene expression datasets and found a substantial overlap with the transcription profile of paraquat-treated animals (Supplementary Fig. 6a, b)[22]. Because the level of antioxidant thiols is lower in DB-fed animals (Fig. 2b), we hypothesized that the level of endogenous ROS under such growth conditions must be elevated, mimicking paraquat treatment. Indeed, dihydroethidium (DHE) staining for ROS in worms reared on DB was higher than on LB (Fig. 4e, f). The DHE signal on DB was sensitive to NAC (Fig. 4e, f), suggesting that the thiols control the level of ROS in worms.

As worms fed LB and DB + NAC exhibited a short lifespan, we propose that only those genes that are similarly regulated by both diets are responsible for accelerated aging. We compared the transcriptomes of worms fed LB or DB + NAC with those fed with DB (Supplementary Fig. 6c, d). According to GO analysis the genes that control lifespan were enriched only among the 124 genes downregulated by both LB and DB + NAC (Supplementary Fig. 6d). A detailed examination of these genes reveals multiple targets of *skn-1* and *daf-16* (Figs. 3b and 4a). Both transcription factors regulate *C. elegans* lifespan and oxidative stress resistance[23–25], indicating that similarly to the effect of NAC,

the downregulation of *skn-1*-mediated response, at least partially, accounts for the shortened lifespan of LB-fed animals.

**GSH restriction extends *C. elegans* lifespan.** As a surplus of reduced thiols accelerates aging, we conjectured that a GSH-restricted diet will extend *C. elegans* lifespan. Previous studies support this hypothesis; a low level of oxidants, as well as the chemicals (diethyl maleate and acetaminophen) that specifically react with and deplete cellular GSH, extend the lifespan[8,26,27]. As reduced GSH is the first line of defense against ROS, its concentration rapidly decreases by oxidant treatment[14], which may, at least partially, explain the anti-aging effect of mild ROS. Furthermore, post-developmental knockdown of *C. elegans* GSH synthase delayed aging[27]. Alternatively, a large body of animal and clinical data demonstrate that severe depletion of intracellular GSH is detrimental[28,29], suggesting that pharmacological application of ROS or temporary disruption of endogenous GSH synthesis, for the purpose of depleting GSH, is hazardous. In contrast, restricting exogenous GSH should not lead to dangerously low levels of endogenous thiols because a sufficient amount of GSH can be synthetized intracellularly.

To test this possibility, we examined the effect of inhibiting GSH import. To cross the cellular membrane, the GSH tripeptide must first be processed[30]. The membrane enzyme γ-glutamyltransferase (γGT) transfers the γ-glutamyl moiety of

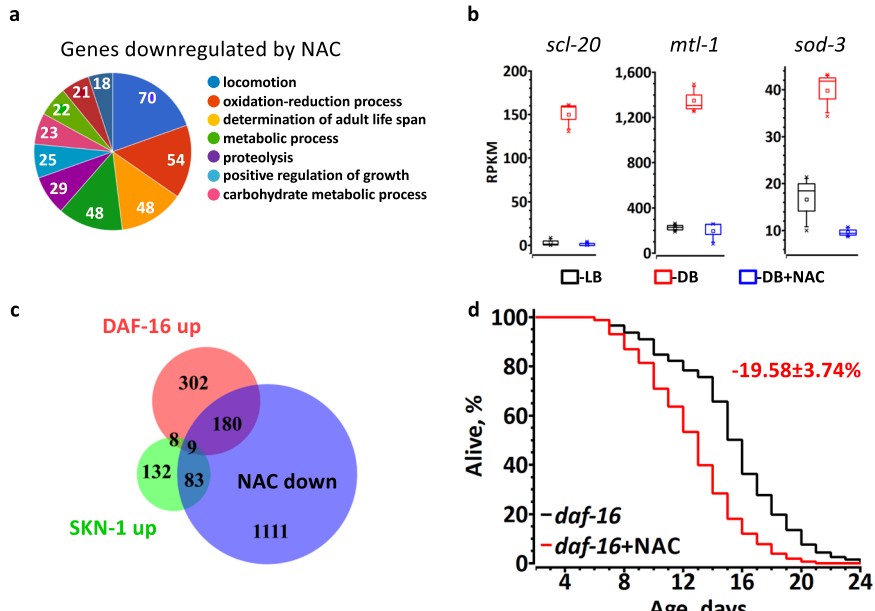

**Fig. 3 NAC downregulates DAF-16 and SKN-1-dependent transcription. a** The most enriched categories (gene ontology) among genes downregulated by NAC. Numbers indicate the number of genes per a category; see Supplementary Data 1 for the full list of differentially regulated genes, and Supplementary Fig. 3 for statistics. **b** Regulation of representative DAF-16 dependent genes by LB, DB or DB + NAC. Graphs show RPKM of three independent experiments from RNA-seq. Box plots indicate median (middle line), 25th, 75th percentile (box) and 5th and 95th percentile (whiskers) as well as maximum, minimum and mean (single points).See Supplementary Data 1 for statistics. **c** Statistically significant overlap between the sets of genes downregulated by NAC and 500 top-ranked DAF-16- (189 genes, RF = 5.6, p value = 4.28e−93) and SKN-1- (92 genes, RF = 5.9, p value = 3.01e−47) dependent genes under normal growth conditions. RF – representation factor is the number of overlapping genes divided by the expected number of overlapping genes drawn from two independent groups and p is normal approximation of hypergeometric probability. **d** NAC (5 mM) shortens the lifespan of *daf-16* worms. Average percentage change ±SD of the lifespan relative to untreated control is indicated in red (n = 304(untreated), 305(5 mM NAC)).

GSH to a free amino acid acceptor and releases the Cys-Gly dipeptide[30]. γGT is a facile pharmacological target because its active site faces the extracellular space. As *C. elegans* contains six putative γGT genes, we decided to use a potent small molecule inhibitor of γGT, acivicin[31], to study the effect of GSH restriction on lifespan.

We first confirmed that acivicin, indeed, does inhibit the accumulation of intracellular GSH in *C. elegans*, when GSH was the only source of exogenous thiols (Supplementary Fig. 7a). However, acivicin decreased the level of intracellular GSH only mildly in worms fed LB (Supplementary Fig. 7b), presumably because they could generate GSH from bacteria-derived Cys and at the expense of other Cys consuming processes in the cell. Indeed, staining with ThioFluor 623 demonstrates that acivicin significantly and dose-dependently decreased the total level of reduced thiols in live animals (Fig. 5a). Remarkably, acivicin treatment beginning at L4 increased the lifespan by ~18% (Fig. 5b). To demonstrate that the anti-aging effect of acivicin was due to the inhibition of intake of GSH-derived thiols, we supplemented DB-fed worms with GSH (Fig. 5c). Predictably, acivicin failed to extend the lifespan of worms fed DB, as this diet is a poor source of GSH (Fig. 2a). Most importantly, acivicin abolished the life-shortening effect of dietary GSH (Fig. 5c), implying that the acivicin effect on aging is specific to GSH import.

**Transcriptional response to GSH restriction and the mechanism of life extension.** To elucidate the anti-aging mechanism of dietary GSH restriction, we studied transcriptome changes in the worm response to acivicin (Fig. 6a, Supplementary Data 2). We found that 23 of 34 acivicin-induced genes and 2 of 5 acivicin-suppressed genes were, respectively, up- and down-regulated in long-lived mitochondrial mutants (Fig. 6a), suggesting that

mitochondria are an effector of GSH restriction. Approximately 38% of acivicin-affected genes are also similarly affected by the *daf-2* mutation (Fig. 6a)[32], implicating the insulin/IGF-signaling pathway in acivicin-induced life extension. Also, many acivicin-modulated genes are similarly affected by mercury or cadmium treatment (Fig. 6a)[33]. As those toxic metals react with and deplete cellular GSH[33,34], this transcriptional response provides additional evidence that acivicin restricts the import of GSH precursors[31].

To further validate our RNA-seq results, we monitored the fluorescence signal in *tbb-6::GFP* worms. Mitochondrial dysfunction upregulates *tbb-6*[35], which occurs via the PMK-3 pathway independently of the classical mitochondrial stress response directed by ATFS-1[35]. Our transcriptomic data show that acivicin activates *tbb-6* by over 7-fold (Fig. 6a). Concordantly, a 24-h acivicin treatment of *tbb-6::GFP* worms increased GFP fluorescence by approximately 6-fold (Fig. 6b). Moreover, NAC, the import of which does not require γGT, increased the intracellular thiols and GSH level (Fig. 2) and decreased *tbb-6::GFP* fluorescence in acivicin-treated worms (Fig. 6b). Additionally, acetaminophen and cadmium, which are known to deplete GSH, induced *tbb-6* (Fig. 6c), indicating that *tbb-6* responds to the decrease of intracellular thiols.

To determine the specific role of GSH in *tbb-6* regulation, we knocked down gamma-glutamylcysteine synthetase (γ-GCS), the first enzyme in GSH biosynthetic pathway. *gcs-1* RNAi did not increase the *tbb-6::GFP* expression (Supplementary Fig. 7c), suggesting that *tbb-6* responds to the low level of cellular thiols in general, not just GSH.

It has been shown that several mitochondrial mutations upregulate *tbb-6* (Fig. 6a) and that *tbb-6* is required for the extended lifespan of such mutant animals[35]. Indeed, mitochondrial uncoupling by dinitrophenol (DNP), which extends mouse

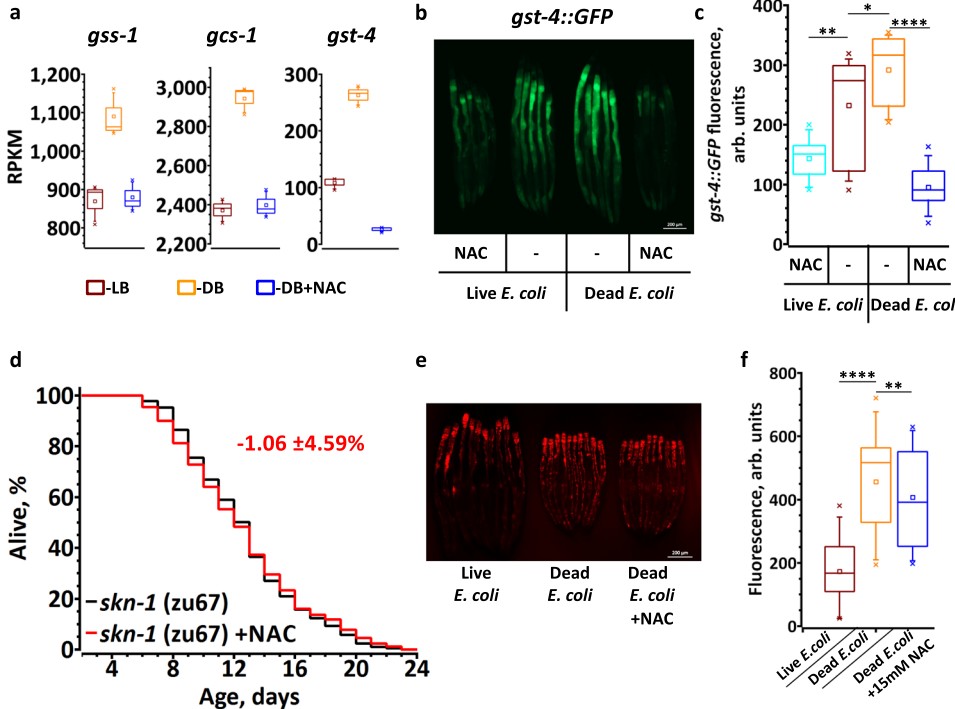

**Fig. 4 Thiols shorten *C. elegans* lifespan by inhibiting SKN-1-dependent transcription. a** Regulation of representative SKN-1 dependent genes by LB, DB or DB + NAC. Graphs show RPKM of three independent experiments from RNA-seq. See Supplementary Data 1 for statistics. **b, c** Representative fluorescent image and quantification demonstrating that *skn-1*-dependent transcription is downregulated by thiols. L4 stage *gst-4::GFP* worms transferred to live or dead bacteria diet ±15 mM NAC and incubated for 5 days prior to imaging. *n* = 60–65 worms over three independent experiments. See also Supplementary Table 5. **d** NAC (5 mM) does not shorten the lifespan of *skn-1* worms. Average percentage change ±SD of the lifespan relative to untreated control is indicated in red (*n* = 188(untreated), 177(5 mM NAC)). **e, f** Representative fluorescent image and quantification demonstrating ROS staining by DHE in live worms. WT animals were reared on LB, DB or DB + 15 mM NAC and stained with DHE at day 3 of adulthood. *n* = 91–94 worms over three independent experiments. See also Supplementary Table 6. In **a, c, f** the box plots indicate median (middle line), 25th, 75th percentile (box) and 5th and 95th percentile (whiskers) as well as maximum, minimum and mean (single points). In all graphs *p* values are: n.s. not significant; *$p < 0.05$; **$p < 0.01$; ***$p < 0.001$; ****$p < 0.0001$; two-tailed *t*-tests.

lifespan[36], also induces *tbb-6* (Fig. 6c). Together, these results argue that mild thiol depletion promotes healthy ROS signaling and triggers the expression of anti-aging genes, such as *tbb-6*. To extend the lifespan, most mitochondrial manipulations should be applied before adulthood[37]. Moreover, recent studies indicate that there is a time window during development for ROS to extend the lifespan[38]. Congruently, we found that acivicin increased mitochondrial ROS production (Fig. 6d) and extended the lifespan almost twice as much (with only a slight delay in development, Supplementary Fig. 10a) if applied to eggs, as compared to stage L4 worms (Fig. 5b).

It has been shown that NAC counteracts the life extension of *nuo-6* and *isp-1* mitochondrial mutants[8], suggesting that the level of GSH in those animals is lower than in wt. Accordingly, feeding worms with DB - a diet deficient in low molecular weight (LMW) thiols, increases the lifespan of wt animals by 58%, but only marginally affected *nuo-6* worms (12% increase, Supplementary Fig. 8a, b).

As the excessive thiols inhibited SKN-1 activity and accelerated *C. elegans* aging (Fig. 4d), we hypothesize that SKN-1 may also be required for acivicin-mediated effects. However, acivicin did not upregulate *gst-4* (Fig. 6a) and increased the lifespan of *skn-1* mutant worms (Fig. 5d), indicating that other anti-aging pathways were activated by the thiol depletion. It has been shown that *daf-16* is required for the anti-aging effect of oxidants[27] that deplete GSH. As indicated earlier, half of the genes affected by acivicin are similarly affected by the *daf-2* mutation (Fig. 6a). Moreover, a large fraction of genes upregulated the most by *daf-*

*16* is suppressed by NAC supplementation (Supplementary Data 1). These results strongly suggest that *daf-16*, perhaps indirectly, senses the redox status of the cell, thereby mediating acivicin-induced life extension. In support of this hypothesis, we found that acivicin failed to increase the lifespan of *daf-16* worms to the extent it did in wt worms (Fig. 5e).

GO analysis of acivicin upregulated genes indicates activation of the unfolded protein response (UPR) (Fig. 7a). As protein homeostasis deteriorates with age, UPR is directly linked to the longer lifespan[6,39]. To validate the GO results, we subjected the animals to treatment with stressors requiring UPR for survival. Although *hsp-4*, a typical marker for ER-UPR, was not upregulated (Supplementary Fig. 9a, b), *C. elegans* pre-treated with acivicin became more resistant to both tunicamycin and heat (Fig. 7c, d). Together these results suggest that the low level of cellular thiols activates mitochondrial signaling and non-canonical UPR to extend *C. elegans* lifespan (Fig. 8). If the lowering of endogenous thiols promoted UPR, we expected that both acivicin and DB diet should similarly elevate stress resistance. Indeed, worms reared on DB were significantly more resistant to heat stress than LB-fed worms (Fig. 7e). Moreover, NAC supplementation, which increases cellular thiols (Fig. 2b), suppressed this resistance (Fig. 7e), demonstrating a connection between the cellular thiol level and UPR.

As acivicin slightly delayed development and significantly decreased the number of progeny (Supplementary Fig. 10a), we asked if its effect on aging could be attributed to the inhibition of germ cell proliferation. Germless *C. elegans* exhibit elevated ROS,

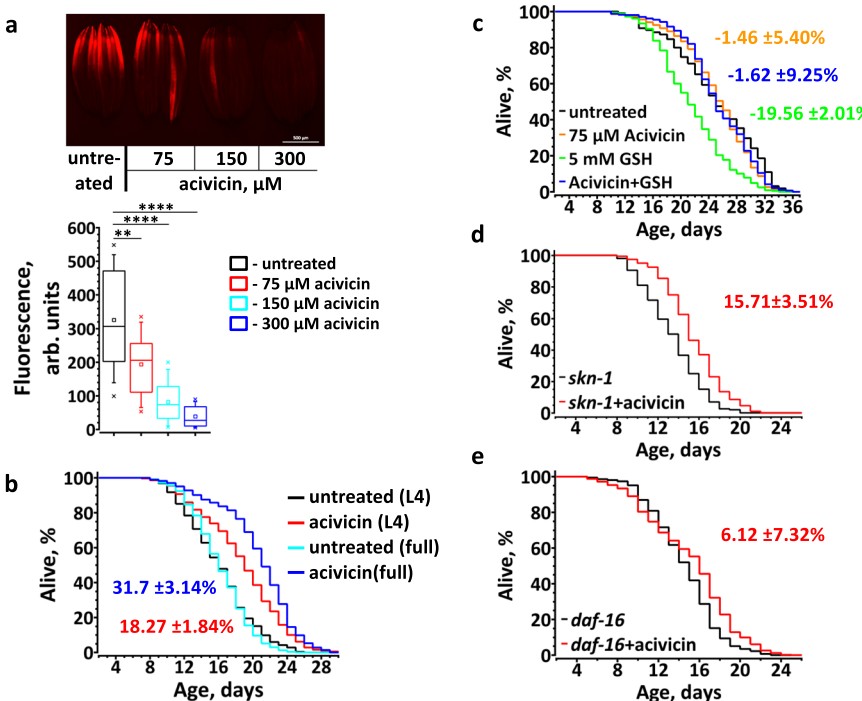

**Fig. 5 Restriction of exogenous GSH increases *C. elegans* lifespan. a** Representative fluorescent image (top) and quantification (bottom) demonstrating total thiols level in live animals exposed to the escalating concentrations of acivicin. L4 stage wt worms were reared on plates supplemented with various concentration of acivicin. At day 3 of adulthood worms were stained with ThioFluor 623. Box plot indicate median (middle line), 25th, 75th percentile (box) and 5th and 95th percentile (whiskers) as well as maximum, minimum and mean (single points). *n* = 68–126 worms over three independent experiments. See also Supplementary Table 7. On the graph *p* values are: **\**p* < 0.01; **\*\*\*\**p* < 0.0001; two-tailed *t*-tests. **b** The γGT inhibitor, acivicin, significantly extends the lifespan of *C. elegans*. Worms were exposed to acivicin either from stage L4 (L4, red trace) or from eggs (full, blue trace). *n* = 256(untreated from L4), 264(acivicin from L4), 198(untreated from eggs), 309(acivicin from eggs). **c** Acivicin extends *C. elegans* lifespan by inhibiting GSH uptake by *C. elegans*. L4 stage worms were transferred to plates with killed bacteria and supplemented with acivicin, GSH, or both (*n* = 154(untreated), 193(acivicin), 191(GSH), 240(acivicin + GSH). **d** Acivicin extends the lifespan of *skn-1* (zu67) worms (*n* = 123(untreated), 146(acivicin)). **e** Acivicin fails to substantially extend the lifespan of *daf-16* worms. Average percentage change ±SD of the lifespan relative to untreated control is indicated in matching color (*n* = 324(untreated), 321(acivicin)).

upregulated mito-UPR, SKN-1 and DAF-16-dependent transcription, and significantly longer lifespan[40,41]. Surprisingly, we found that the extended lifespan of germless *glp-1* worms was shortened by acivicin (Supplementary Fig. 10b). Acivicin increased mitochondrial ROS (Fig. 6d), but did not induce the mito-UPR marker *hsp-6* (Supplementary Fig. 9c, d), arguing that acivicin and *glp-1* induce different signaling. Moreover, in contrast to *glp-1* mutants[41,42], acivicin extended the lifespan of *skn-1* animals (Fig. 5d). We also found that thiols have been severely depleted in *glp-1* worms and could not be further reduced by acivicin (Supplementary Fig. 10c, d). Taken together, these results suggest that acivicin does not inhibit germ line signaling per se, but has a common DAF-16-dependent downstream target(s) that regulate lifespan.

**GSH restriction induces ER-UPR in human dermal fibroblasts.** We next examined whether the transcriptional response to dietary thiol restriction was conserved between worms and human cells. Treatment of human dermal fibroblasts with acivicin, which inhibits γ-GT and restricts GSH import to mammalian cells[43], as it does in worms, results in a substantial transcriptomic change (Supplementary Data 3). Gamma-glutamyltransferase 1 (GGT1) gene expression was increased approximately 9 fold (Supplementary Data 3), demonstrating its negative regulation by cellular thiols. GO analysis indicates that, similarly to *C. elegans*, ER UPR is among the most upregulated categories in human cells (Fig. 7b). The master regulator of ER-UPR, XBP-1, was upregulated 2-fold along with several other proteins (e.g., DNAJB9 and

HSPA5T - ~5.5 fold) typically induced by ER stress (Supplementary Data 3).

To further substantiate these results, we examined whether acivicin renders fibroblasts more resistant to ER stress. Indeed, acivicin-treated cells survived the treatment with tunicamycin and heat shock substantially better (Fig. 7f, g and Supplementary Fig. 11c, d). Note that the increasing concentration of acivicin progressively inhibited cells proliferation, without killing them (Supplementary Fig. 11a, b), indicating that exogenous GSH is important for rapid cell growth, but not for survival.

It has been shown that the DNAJB9 and HSPA5T expression was HSF1-independent[44]. Accordingly, we found that knocking down HSF1 did not compromise acivicin-induced tunicamycin stress resistance (Supplementary Fig. 11e), indicating that acivicin specifically induced ER-UPR.

Together these results illuminate an evolutionary conserved pathway that links low cellular thiols to the activation of the ER stress response and geroprotection.

## Discussion
Antioxidant supplements, such as LMW thiols, can protect against exogenous ROS toxicity[1,45]. However, endogenously generated ROS act as ubiquitous and important signaling molecules that regulate cellular homeostasis, differentiation, proliferation, repair, and aging[45,46]. Therefore, it is not surprising that popular antioxidant supplements, including NAC, GSH, vitamin E and C, fail to show any significant health benefits in most long-term well-controlled clinical trials[45]. In particular,

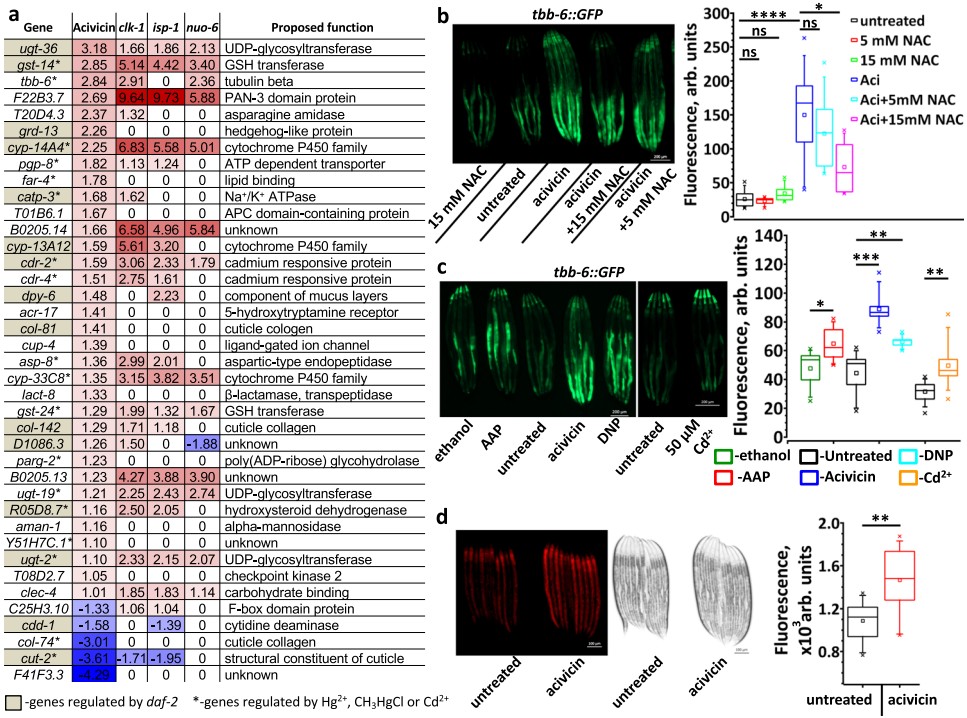

**Fig. 6 GSH restriction upregulates mitochondrial ROS signaling in *C. elegans*. a** Genes induced by acivicin and by life-extending mitochondrial mutations overlap significantly. Numbers represent the expression $\log_2$ fold change compared to wt or untreated worms. See full list of differentially regulated genes in Supplementary Data 2. Data for gene expression in *daf-2*, *clk-1*, *isp-1* and *nuo-6* are from[32]. Up- and down-regulated genes are indicated, respectively, in red and blue. Genes which are regulated in *daf-2* worms are highlighted in gray. *-genes regulated by $Hg^{2+}$, $CH_3HgCl$ or $Cd^{2+}$. Statistical significance for the overlap between sets upregulated more than two folds by: acivicin (34 genes) and *nuo-6* (1285 genes) – 22 genes overlap, RF = 10.3, *p* value = 7.93e−19; acivicin and *clk-1* (328 genes) – 13 genes overlap, RF = 23.9, *p* value = 2.48e−15; acivicin and *isp-1* (609 genes) – 19 genes overlap, RF = 18.8, *p* value = 9.01e−21. RF – representation factor is the number of overlapping genes divided by the expected number of overlapping genes and *p* is normal approximation of hypergeometric probability. **b, c** Representative fluorescent images (left panels) and quantifications (right panels) demonstrating the *tbb-6* counter-regulation by acivicin (*n* = 70) and NAC (*n* = 30–55) (**b**) and the *tbb-6* induction by acetaminophen (AAPh, 20 mM, *n* = 35), acivicin (75 μM, *n* = 35), dinitrophenol (DNP, 100 μM, *n* = 35), and cadmium ($Cd^{2+}$, 50 μM, *n* = 65) (**c**). L4 stage worms were transferred to plates supplemented with chemicals as indicated, and incubated for 24 h at 20 °C before imaging. *n* = 30–70 worms over three independent experiments. See also Supplementary Tables 8 and 9. **d** Representative fluorescent and bright light images (left panels) and quantifications (right panel) demonstrating mitochondrial ROS staining by MitoTracker CM-H$_2$X in control and acivicin-treated worms (*n* = 79). Worms developed till stage L3 on control and acivicin supplemented plates. See also Supplementary Table 10. In **b, c** and **d** the box plots indicate median (middle line), 25th, 75th percentile (box) and 5th and 95th percentile (whiskers) as well as maximum, minimum and mean (single points). In all graphs *p* values are: n.s. not significant; *$p < 0.05$; **$p < 0.01$; ***$p < 0.001$; ****$p < 0.0001$; two-tailed *t*-tests.

several such clinical trials have been terminated prematurely due to accelerated cancer progression and higher incidence of cancer-related mortality[45,47,48].

As mitochondria are the primary source of ROS, their function could be compromised by excessive antioxidants. Indeed, antioxidants, including vitamins C and E, blunt the beneficial effects of exercise, such as mitochondrial biogenesis[49–51]. In brain endothelial cells, antioxidants negatively impact mitochondrial function[52,53]. This growing body of evidence calls into question the popular belief about health benefits of chronic consumption of antioxidants and necessitates a better mechanistic understanding of their action at the organismal level.

To this end, we explored the life-long effects of common LMW thiol antioxidants, NAC and GSH, on *C. elegans* aging. Some previous reports, which detected higher ROS in old animals, suggested that antioxidants may increase the lifespan[10]. In contrast, we observed that LMW thiols accelerate *C. elegans* aging in a dose-dependent manner (Fig. 1a–c). Notably, NAC supplementation even to older animals decreased the lifespan (Fig. 1d), indicating that scavenging naturally occurring ROS is harmful at any age. Earlier attempts to establish the effect of NAC on the *C. elegans* lifespan have been inconclusive, ranging from lengthening

to shortening of the lifespan or to having no effect[8,15,41,54]. Redox active NAC is prone to oxidation and readily metabolized by bacteria, which alters its bioavailability. Moreover, NAC is a stronger acid than Cys and GSH, causing media acidification, which itself influences the *C. elegans* lifespan[55]. Finally, occasionally used additives, such as FUDR or DMSO[54], counteracted the negative effect of NAC (Supplementary Fig. 1b). Therefore, we supplemented pH-adjusted NAC on the day of the experiment and consistently observed shortening of the lifespan (Fig. 1).

We found that dietary NAC downregulates many genes, some of which are positively controlled by DAF-16 and SKN-1 (Figs. 3b, 4a, b, and Supplementary Data 1). Both of these transcription factors participate in the oxidative stress response and become activated by redox switches. Oxidation of IMB-2 Cys residues promotes DAF-16/FOXO nuclear translocation and DNA binding[19]. Oxidants also promote SKN-1 nuclear localization via the p38 mitogen-activated protein kinase (MAPK) pathway[18,56]. Under non-stressed conditions, the WDR-23 adapter protein targets SKN-1 for degradation[20]. It was hypothesized that upon oxidative stress, Cys-rich WDR-23 can dissociate from SKN-1, thereby promoting activation of SKN-1-mediated transcription[20,23]. Our results indicate that the

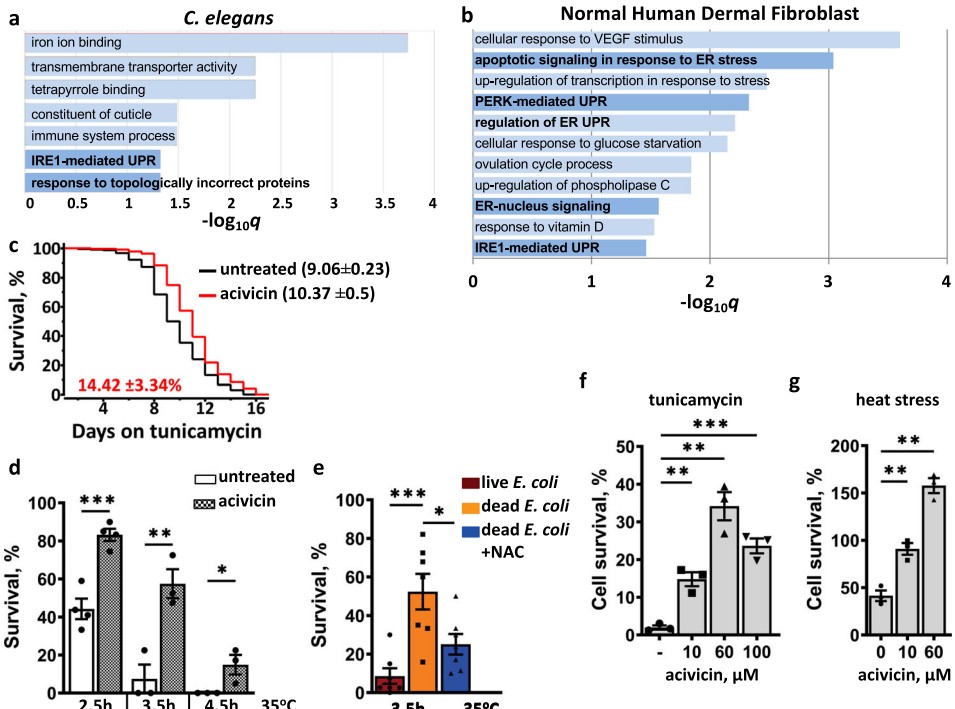

**Fig. 7 Evolutionary conservation of the response to dietary thiols restriction.** UPR is the highly enriched (gene ontology) category among the genes upregulated by acivicin in *C. elegans* (>0.5 log₂ fold) (**a**) and in human dermal fibroblasts (>1 log₂ fold) (**b**); see also Supplementary Data 2 and 3. Worms pretreated with acivicin are resistant to tunicamycin (**c**) and heat (**d**). WT worms were exposed to 75 μM acivicin from eggs until day 2 of adulthood and then subjected to proteotoxic stress (see Experimental procedures). The graph represents the composite of three independent experiments. In **c**, the average percentage survival change ±SD relative to untreated control is indicated in red ($n = 150$). In **d**, the graph represents an average ±SEM from at least three independent experiments ($n = 109$–150 worms). **e** Worms reared on dead *E. coli* diet are resistant to heat stress. WT worms reared on live, dead or dead+NAC *E. coli* diet until day 2 of adulthood and then subjected to heat stress (35 °C, 3.5 h). The graph represents an average ±SEM from seven independent experiments ($n = 201$–278 worms). See also Supplementary Table 11. Normal human dermal fibroblasts pretreated with acivicin are resistant to tunicamycin (**f**) and heat (**g**). Cells were pretreated with indicated concentration of acivicin for 24 h and then exposed to 10 μg ml⁻¹ tunicamycin for 48 h (**f**) or heat shocked at 46 °C for 3 h (**g**). Survival was assessed 24 h later. Three independent experiments were performed and percent survival ±SEM was calculated against unstressed control supplemented with the same concentration of acivicin. See also Supplementary Fig. 11c, d. In all graphs *p* values are: n.s. not significant; $*p < 0.05$; $**p < 0.01$; $***p < 0.001$; $****p < 0.0001$; two-tailed *t*-tests.

inhibition of SKN-1 is one of the major causes of accelerated aging by exogenous thiols, as NAC shortened the lifespan of *daf-16* animals, but failed to do so for *skn-1* mutants (Figs. 3d and 4d). Curiously, NAC accelerated the aging of *daf-16* animals almost twice as much as that of wt animals (compare Figs. 1a and 3d), suggesting that SKN-1 compensated for some critical functions in the absence of DAF-16.

It has been shown that SKN-1 activation is required for the higher expression of GSH biosynthetic genes and elevated GSH[57]. However, as GSH and NAC do not extend *C. elegans* lifespan, this function of SKN-1 is an unlikely contributor to longevity.

Notably, our results show that excessive antioxidant thiols originate from the host microbiota. *E. coli*, which is common food for laboratory *C. elegans*, is an enormous reservoir of reduced LMW thiols. With millimolar levels of reduced GSH[12], these bacteria are akin to an antioxidant supplement taken by *C. elegans* for life. Adjusted to the total protein level, the concentration of *E. coli* GSH is several times higher as compared with that in *C. elegans* (Fig. 2a, b). Moreover, GSH biosynthetic genes are downregulated in worms fed LB and DB + NAC compared to worms fed DB, clearly indicating an adaptation to high exogenous thiols (Fig. 4a). Several lines of evidence support our conclusion that reduced thiols from live bacteria shorten the worm lifespan: (i) *C. elegans* live significantly longer when fed *B. subtilis*[58], which do not make GSH and contain ten times less (<1 mM) LMW thiols than do *E. coli*[59,60]; (ii) Killed *E. coli* have a lower content of

reduced thiols, resulting in an increased *C. elegans* lifespan (Fig. 1)[11]. (iii) A specific thiol oxidizing agent, diamide, lengthens the lifespan of LB- fed *C. elegans*[14], but does the opposite to DB-fed *C. elegans*, indicating that oxidation of bacterial thiols is the major benefit of diamide (Supplementary Fig. 1d, e). Taken together, these results argue that thiol-rich bacteria are harmful for their host, as the excess of antioxidants they provide inhibits healthy ROS signaling and suppresses anti-aging transcription, which is mediated by SKN-1 and, possibly, other transcription factors (Fig. 4).

Our results suggest that animals need to curb their antioxidant thiol intake to extend their lifespan, as opposed to supplementing their diet with antioxidants. Oxidative stress induced by chemicals (e.g., paraquat, arsenite, acetaminophen), or by genetic manipulations, uniformly increases the *C. elegans* lifespan[8,26,27,41,61,62]. However, both approaches are impractical, as they deteriorate the endogenous capacity of the organism to withstand acute stress. Accordingly, restricting exogenous thiols provides a safer alternative for mobilizing cellular defense programs. All animal cells can synthetize a sufficient amount of GSH to protect against sudden ROS fluctuations and to maintain normal cellular redox status. We show that restricting exogenous GSH by acivicin, while substantially decreases the overall thiol level and extending the lifespan, reduces endogenous GSH only mildly (Fig. 5a, b and Supplementary Fig. 7b). Likewise, acivicin lowers intracellular Cys, but not GSH, in human cells, indicating

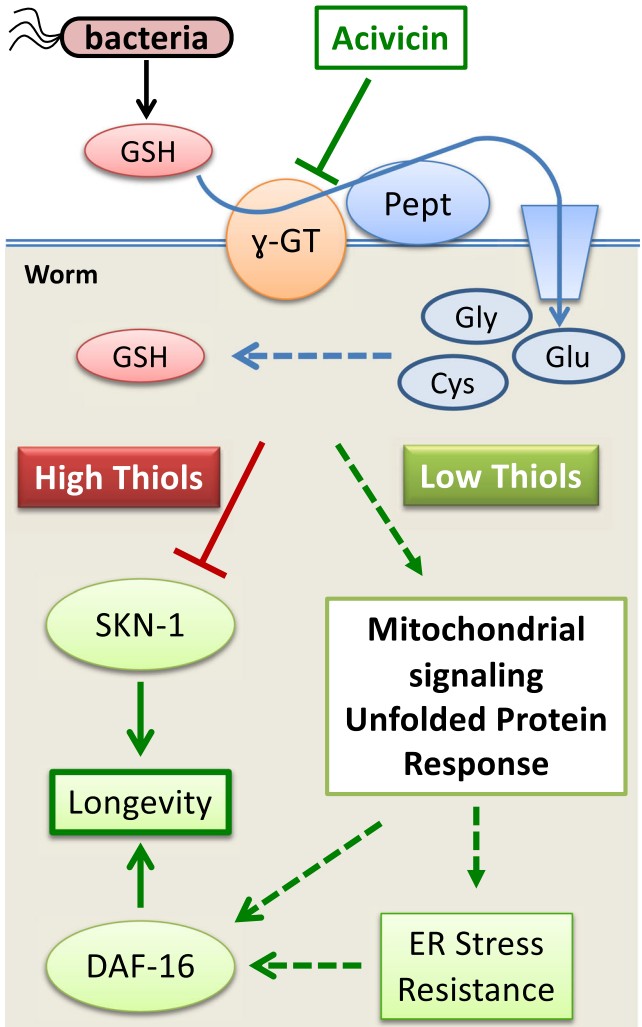

**Fig. 8 The proposed mechanism of the *C. elegans* response to exogenous thiols.** An excessive amount of GSH supplied by an *E. coli* diet leads to accumulation of a higher level of endogenous thiols (including GSH and Cys), which perturbs global gene expression, including the inhibition of SKN-1-mediated transcription, and accelerates aging. Acivicin restricts dietary GSH intake, favors an increased ROS production in mitochondria, which triggers the UPR-like response, enhances proteotoxic stress resistance, ultimately leading to an extended lifespan.

that human cells and worms are capable of maintaining GSH at the expense of other Cys-utilizing processes[43]. Consistent with our findings, it was shown that diets low in sulfur containing amino acids prevent various pathologies and extend the lifespan of several model organisms[63].

Analysis of gene expression in acivicin-treated worms shows that more than half of the affected genes are similarly affected by mitochondrial mutations that extend the lifespan[32] (Fig. 6a). One of the most upregulated genes, *tbb-6*, is induced by the specific p38 MAP kinase-signaling cascade (PMK-3) in response to disruption of the electron transfer chain (ETC) in mitochondria[35]. Although *tbb-6* itself has a minor effect on the lifespan, knocking out its regulator, *pmk-3*, attenuates the longevity phenotype of several mitochondrial mutants[35]. Our data demonstrate that a common mitochondrial uncoupler dinitrophenol, which extends the mouse lifespan[36], induces *tbb-6* expression similarly as does acivicin, further indicating that both chemicals target ETC (Fig. 5).

Previous work has shown that the *tbb-6* expression was independent of ATFS-1 or SKN-1[35]. ATFS-1, a key regulator of mitochondrial UPR, increases the expression of SKN-1 and the chaperones, such as *hsp-6* and *hsp-60*, in response to mitochondrial disturbances[64,65]. As acivicin does not induce *hsp-6* (Supplementary Fig. 9c, d) and does extend the lifespan of *skn-1* mutant worms (Fig. 5d), it should not act via ATFS-1. Indeed, acivicin deregulates a much smaller set of genes comparing to the life-prolonging mitochondrial mutation (Fig. 6a)[32].

We also found that *tbb-6* is upregulated in response to GSH-depleting chemicals (Fig. 6c), suggesting that it is induced by the lowered thiols and/or high ROS (Fig. 6d). Other acivicin-regulated genes are similarly affected by toxic metals ($Cd^{2+}$ and $Hg^{2+}$) (Fig. 6a), which are known to deplete GSH[33,34]. The well-established anti-aging mutants, *isp-1*, *nuo-6*, *glp-1* and *daf-2*, all have elevated mitochondrial superoxide[8,41]. NAC, at least partially, suppresses the life extension phenotype of these animals, apparently by scavenging ROS[8,41]. Consistently, restricting GSH intake by feeding worms with DB failed to prolong the lifespan of *nuo-6* animals to the same extent as the wt (Supplementary Fig. 8a, b). Moreover, we found that *glp-1* worms have the level of cellular thiols greatly diminished (Supplementary Fig. 10c, d). Elevated $H_2S$ production in *glp-1* could be the reason for this phenotype[41]. Low thiols and high ROS in *glp-1* animals may explain why they no longer benefit from ROS[41] or from the further GSH depletion by acivicin (Supplementary Fig. 10b–d). Together these results argue that it has to be an optimal balance between endogenous thiols and ROS to lengthen the lifespan.

Among the genes induced by acivicin, the UPR genes were enriched in both worms and human cells (Fig. 7a, b). Accordingly, the low level of endogenous thiols resulted in stronger resistance to proteotoxic stress (Fig. 7c–g), which was HSF1-independent in human cells (Supplementary Fig. 11e). Similarly, in *C. elegans*, almost none of the heat shock proteins were upregulated by acivicin (Supplementary Data 2 and 3). For example, *hsp-4*, the most studied ER chaperone in *C. elegans*, was not induced (Supplementary Fig. 9a, b and Supplementary Data 2). Such a response is reminiscent of that of the *daf-2* mutant, where *daf-16* and *xbp-1* act to maintain ER homeostasis (and increasing the lifespan) without inducing HSPs[66]; they act by promoting degradation of misfolded proteins and reducing the translation rate. Accordingly, acivicin-mediated life extension also requires *daf-16* (Fig. 5e).

To summarize, we demonstrate the detrimental effect of exogenous antioxidant thiols on the *C. elegans* lifespan and suggest that the molecular mechanism by which it occurs relies predominantly on the inhibition SKN-1-mediated transcription. Our results argue that inversely coupled low thiols/high ROS in mitochondria may trigger UPR, the elevated stress resistance, and increased lifespan (Fig. 8). Considering the high evolutionary conservation of the biochemical pathways involved (Fig. 7), our findings advocate against the long-term consumption of antioxidant supplements, at least until their life-long effects have been thoroughly investigated in mammals. We further advocate for future studies in mammals to investigate the pharmacological restriction of exogenous thiols as a potentially safe and potent action to promote healthy ROS signaling and counteract aging.

## Methods

**Nematodes**. Wild-type *C. elegans* (N2), CF1038 (*daf-16(mu86) I*), CL2166 (*dvIs19* [(pAF15)*gst-4p*::GFP::NLS] *III*), EU1 (*skn-1(zu67) IV*/nT1 [*unc-?(n754) let-?*] (IV; V)), SLR115 (*dvIs67* [*tbb-6p*::GFP+*myo-3p*::dsRed]), SJ4005 (*zcIs4* [*hsp-4*::GFP] V), SJ4100 (*zcIs13*[*hsp-6*::GFP]), CF1903 (*glp-1*(e2144) III), GR1373 (*eri-1*(mg366) IV) and MQ1333 (*nuo-6*(qm200) I) strains were obtained from the *Caenorhabditis* Genetics Center and handled according to standard methods[67]. Strains were grown on NGM agar plates at 20 °C. *E. coli* OP50 bacteria were grown overnight in Luria-

Bertani broth. *C. elegans* growth plates with live bacteria (LB) were made by spreading 50 μl of a 10X concentrated overnight bacterial culture atop NGM agar plates. For dead bacteria (DB), NGM was supplemented with bactericidal antibiotic mix (50 μg ml$^{-1}$ kanamycin and 100 μg ml$^{-1}$ carbenicillin). Plates were incubated for at least 1 h at 20 °C before worms were transferred to them. Heat killed bacteria were prepared by incubation of a 20X concentrated overnight culture for 1 h at 65 °C.

On the day of the experiment additives were evenly distributed on wet NGM plates to a final concentration of: 5 or 15 mM NAC, 5 mM GSH, 2.5 or 5 mM diamide, 20 mM acetaminophen, 10 or 50 μM Cd$^{2+}$, 100 μM dinitrophenol, or 75 μM acivicin plates were dried for 30–60 min before seeding with bacteria. Chemicals were dissolved in dH$_2$O with the exception that acetaminophen was dissolved in ethanol. NAC stock solution, 0.8 M pH = 5.5 (adjusted to with NaOH), was stored in small aliquots frozen at −20 °C.

For RNAi experiments, eggs were isolated by treating adult hermaphrodites with alkaline hypochlorite and allowed to develop and grow for two generations on specific RNAi expressing bacterial strains before being used or lifespan analyses. *E. coli* HT115 strains harboring plasmid expressing double strand RNAi against *C. elegans wdr-23* gene was purchased from Thermo Scientific collection and single-colony isolates were purified and sequenced to demonstrate the presence of the correct insert. Overnight cultures of *E. coli* HT115 bacteria harboring RNAi expressing plasmid or empty plasmid vector control (pL4440) were grown in Luria-Bertani broth with 100 μg ml$^{-1}$ of carbenicillin, concentrated 6 times and 50 μl were spread atop NGM agar plates supplemented with 100 μg ml$^{-1}$ carbenicillin and 1 mM IPTG. Seeded plates were incubated for at least 1 h at 20 °C or 25 °C before worms were transferred onto them.

**Human cells.** Normal Human dermal fibroblasts (ATCC, PCS-201-012) were grown in Dulbecco's modified Eagle's medium (DMEM) supplemented with 10% (v/v) fetal bovine serum, 10 mM glutamine, pH 7.4. Cells were grown to 70–80% confluence in 100-mm dishes and then treated with 100 μM Acivicin for 24 h.

**Lifespan analyses.** Lifespans were monitored at 20 °C or 25 °C (*wdr-23* RNAi only) as described previously[68,69]. Details regarding repeat experiments and amounts of animals used for experiments are summarized in Supplementary Table 2. In all cases, stage L4 worms were used at t = 0 for lifespan analyses and worms were transferred every 2–3 days to new agar plates. Worms were judged to be dead when they ceased pharyngeal pumping and did not respond to prodding with a platinum wire. Worms with internal hatching were removed from the plates and not included in lifespan calculations. Data were analyzed and Boltzmann sigmoid survival curves generated using the SciDAVis statistical analysis software package. Mean lifespans were compared in Microsoft Excel using the Student *t* test, assuming two-tailed distribution and paired. All lifespan plots represent the composites of all independent experiments tabulated in Supplementary Table 2. Mean percentage change ±SD of life span after treatment relative to untreated control is indicated in each graph in same color as a curve.

For experiments in the *eri-1* background, which allows the RNAi in neuronal tissues, L4 stage worms were shifted to 25 °C and kept for their reminding life span. *glp-1* worms were handled as described previously[40,41]. Briefly, worms were maintained at 20 °C. Adult worms were let lay eggs and plates transferred to 25 °C. L4 stage worms picked and transferred to 20 °C for the rest of their life.

**Paraquat resistance assay.** Adult wt worms were allowed to lay eggs on LB or DB NGM plates. LB plates were made by spotting 50 μl of a 10X concentrated overnight culture of *E. coli* OP50 atop NGM plates. For DB, NGM plates were supplemented with bactericidal antibiotic mix (50 μg ml$^{-1}$ kanamycin and 100 μg ml$^{-1}$ carbenicillin). After worms reached stage L4, they were transferred to fresh LB or DB plates+/− 15 mM NAC. After 24 h at 20 °C, worms were washed from the plates, 3 times with M9 buffer, and then exposed to 150 mM paraquat in M9 buffer (~40 animals per each experimental condition). After a 1-h incubation at 20 °C with agitation worms were washed 4 times in M9 buffer to remove traces of paraquat and spotted on fresh NGM plates with 40 μM FUDR and seeded with bacteria. Plates were incubated at 20 °C, live worms were counted and the survival rate calculated as a ratio of those alive to the total number of worms transferred to the plate after oxidant treatment. All experiments were repeated at least three times and the average ±SD presented in the Fig. 2c and Supplementary Fig. 1c.

**Determination of reduced thiol content.** Total cellular reduced thiols were quantified by reaction with DTNB (5,5′-dithiobis-(2-nitrobenzoic) acid)[70]. Worms were allowed to develop and grow at 20 °C until stage L4, and then transferred to LB, DB or DB + 15 mM NAC agar plates. For each experimental condition ~100 worms were collected in micro-centrifugal tubes and washed quickly 3 times with M9 buffer. After removing most of the liquid the worms were flash frozen in liquid nitrogen. For analysis, 50 μl of 40 mM HEPES pH=7.4, 1 mM EDTA and 6 M Guanidine HCl buffer was added to the tubes with worms, frozen animals grinded with a disposable pestle and lysed by two freeze thaw cycles. The lysate was diluted with 125 μl of 40 mM HEPES pH=7.4, 1 mM EDTA and insoluble materials were removed by filtration with COSTAR$^R$ Spin-X centrifuge tube filter. DTNB was

added to an aliquot of the flow-through to the final concentration of 2 mM and samples incubated for 5 min before the absorbance at 412 nm was determined. The concentration of thiols in the samples was calculated according to a standard curve generated by reaction of DTNB with glutathione and then adjusted to a protein concentration (BCA assay) in the lysates.

To quantify the thiol content in bacteria, an overnight culture of *E. coli* OP50 was concentrated 20X and 50 μl spotted on an NGM plate with or without 100 μg ml$^{-1}$ carbenicillin and 50 μg ml$^{-1}$ kanamycin. Three days later bacteria were washed from the plates with 200 mM MES, 50 mM phosphate and 1 mM EDTA, pH=6 buffer. Bacteria were treated with lysozyme (125 μg ml$^{-1}$ final) for 10 min on ice, sonicated, and the lysate clarified by centrifugation. The supernatant was used for reaction with DTNB as described above.

**Determination of total GSH content.** Glutathione Assay Kit (Cayman chemical, 703002) was used to determine GSH concentration in worm lysates. About 200 worms were picked, washed with 200 mM MES, 50 mM phosphate and 1 mM EDTA, pH=6 buffer and flash frozen in liquid nitrogen. Pellets were grinded with pestles in 100 μl of the same buffer, cells were lysed by two freeze-thaw cycles and insoluble material was removed by filtration in a COSTAR$^R$ Spin-X centrifuge tube filter. GSH was measured according to the kit manual and then adjusted to the protein concentration (BCA assay) in the lysates.

**ROS staining.** Wt animals reared on LB or DB. l4 stage animals were transferred to LB, DB or DB + 15 mM NAC plates and incubated at 20 °C. At day 3 of adulthood worms were picked, washed with PBS and stained with 5 μM of dihydroethidium (DHE, Invitrogen) in PBS for 45 min at 20 °C with agitation. Stained worms washed extensively, anesthetized in a drop of 2% sodium azide and images were captured immediately using a Zeiss AxioZoom v16 microscope equipped for fluorescence illumination. Fluorescence intensity was quantified using the Zeiss ZEN software package. As worms have different size we tightly traced groups of worms around and read the mean intensity.

Mitochondrial ROS were stained with MitoTracker Red CM-H$_2$X (Invitrogen) as described in[71]. Briefly, wt nematodes were incubated on control and 75 μM acivicin supplemented plates from eggs till stage L3-L4, picked, washed and transferred to incubation plates. To make incubation plates 500 μl heat-inactivated OP50 (65 °C, 30 min) was mixed with 20 μl MitoTracker Red CM-H$_2$X stock solution (0.5 mM) and 50 μl spotted on a NGM agar plates. Worms were incubated on MitoTracker Red CM-H$_2$X plates for 2 h at 20 °C. To remove excessive dye from the gut, worms were washed and transferred to NGM agar plates with or without acivicin. After 1 h at 20 °C worms were picked washed and anesthetized in a drop of levamisole and images were captured immediately using a Zeiss AxioZoom v16 microscope equipped for fluorescence illumination. Fluorescence intensity was quantified using the Zeiss ZEN software package.

**In vivo thiol staining.** Stock solution of ThioFluor 623 (10 mM) was prepared in DMSO and stored at −80 °C and diluted in M9 immediately before the use. Worms picked, washed twice in M9 buffer and incubated in 1 ml of 25 μM ThioFluor 623 in M9 buffer. After 15 min incubation at room temperature worms were washed extensively, anesthetized in a drop of 2% sodium azide and images were captured immediately using a Zeiss AxioZoom v16 microscope equipped for fluorescence illumination. Fluorescence intensity was quantified using the Zeiss ZEN software package.

**Fluorescent reporter assays.** Worms expressing GFP under control of *tbb-6* (SLR115) and *gst-4* (CL2166) were fed on NGM plates seeded with *E.coli* OP50. One- or two-day old adult worms were anesthetized in a drop of 2% sodium azide and images were captured immediately using a Zeiss AxioZoom v16 microscope equipped for fluorescence illumination. Fluorescence intensity was quantified using the Zeiss ZEN software package.

**Development and progeny production.** Gravid adults were placed on control and acivicin (75 μM)-supplemented NGM plates seeded with 50 μl of *E. coli* OP50. Worms were allowed to lay eggs for 2 h and then removed from the plates. Worms were scored for a life stage 65 h later. The experiment was done in triplicate and average percentage presented in a graph. To count a progeny production, worms were exposed to 75 μM acivicin from eggs throughout the experiment. Stage L4 worms (n = 19 for untreated control and 9 for acivicin) were placed on individual plates and the amount of eggs laid counted. The graph shows the averages ±SEM.

**Tunicamycin and heat tress resistance in *C. elegans*.** Gravid adults were allowed to lay eggs on NGM plates with or without 75 μM acivicin and spotted with *E. coli* OP50. One hour later, adults were removed and worms were allowed develop and grow at 20 °C until they reached stage L4. Synchronized L4 stage worms were picked and incubated with or without acivicin until day two of adulthood (A2). For heat shock, ~50 A2 worms were transferred on new NGM plates seeded with *E. coli* OP50 without acivicin and incubated for 2.5, 3.5 and 4.5 h at 35 °C. Plates were transferred to 20 °C and surviving worms were counted 48 h later. All experiments

were repeated three times and the average ±SEM percent survival presented in the Fig. 7. For tunicamycin resistance ~80 A2 worms were transferred to NGM plates with 50 μg ml⁻¹ tunicamycin seeded with *E. coli* OP50 and live worms were counted every day. Worms were re-transferred to new tunicamycin plates every 4 days. The experiments were repeated three times and the average ±SD days survival on tunicamycin presented in the Fig. 7.

**Cell culture, RNA-seq and stress resistance**. For the total transcriptome analysis, three biological replicates of cells were treated with 100 μM acivicin for 24 h and harvested in Trizol reagent. Total RNA was isolated by phenol/chloroform extraction and isopropanol precipitation. mRNA was purified by NEBNext® Poly (A) mRNA Magnetic Isolation Module (NEB E7490S) from DNAse treated total RNA. RNA-seq libraries were prepared using NEBNext® Ultra™ RNA Library Prep Kit for Illumina® (NEB E7530S).

To determine acivicin toxicity normal human dermal fibroblasts were treated with a range of concentrations of acivicin for 24 or 48 h and apoptosis was measured by Caspase-Glo 3/7 Assay System (Promega).

To study tunicamycin resistance cells were treated with acivicin for 24 h and then media replaced with normal growth medium. Control and acivicin-treated cells were subjected to 10 μg ml⁻¹ tunicamycin for 48 h. Cells were washed to remove detached cells, stained with a live cell dye (Molecular Probes R37609) and then subjected to fluorescent imaging using an EVOS system and Image J for quantification.

For heat stress resistance cells were treated with acivicin for 24 h and then media replaced with fresh growth medium. Control and acivicin-treated cells were subjected to heat shock at 45 °C for 3 h and then transferred to 37 °C for 24 h. Cells were washed to remove detached cells, stained with a live cell dye (Molecular Probes R37609) and then subjected to fluorescent imaging using an EVOS system and Image J for quantification.

Hsf1 was knockdown by siRNA. Two rounds of transfection were performed 24 h apart. Cells were then treated with acivicin for 24 h, as indicated in the Fig. 7. Acivicin containing media was replaced with normal growth medium. Control and acivicin-treated cells were subjected to 10 μg/ml Tunicamycin for 48 h. siRNA sequences to knock down Hsf1 are:

Antisense1: rCrUrUrGrArUrGrUrUrCrUrCrArArGrGrArGrCrUrGrCrUrCr CrUrG

Sense1: rGrGrArGrCrArGrCrUrCrCrUrUrGrArGrArArCrArUrCrAAG

Antisense2: rArArGrUrGrGrUrCrArCrUrGrArGrCrUrCrArUrUrCrUrUrGr UrCrC

Sense2: rArCrArArGrArArUrGrArGrCrUrCrArGrUrGrArCrCrArCTT

Negative control siRNA - IDT catalog no. 51-01-14-04

**Sequencing and differential expression analyses**. To study transcriptional response to NAC wt worms were allowed to develop and grow on LB, DB or DB + 15 mM NAC agar plates at 20 °C until day 8 of adulthood. Heat-inactivated bacteria were used to prepare DB agar plates. To avoid extensive internal hatching 40 μM FUDR was added after the worms reached L4 stage.

To study the transcriptional response to acivicin, wt worms were allowed to develop and grow on LB agar plates at 20 °C until they reached stage L4, and then transferred to LB agar plates with or without 75 μM acivicin and incubated for 24 h at 20 °C.

About 200 worms were collected, washed in M9-buffer, and total RNA isolated as described in[72]. mRNA was purified by NEBNext® Poly(A) mRNA Magnetic Isolation Module (NEB E7490S) from DNAse treated total RNA. A NEB Next® Ultra Library Preparation Kit (NEB E7530S) was used to prepare 1 μg of total RNA for RNA-seq. Three independent biological replicates were used for each experimental condition and six replicates for acivicin-treated worms. The libraries were sequenced using Illumina NextSeq 500 instrument in a paired-end 2×75 cycles setup. The reads were aligned against Wbcel235 *C. elegans* genome assembly using Hisat2 version 2.1.0[73]. The number of reads in annotated genes was counted using htseq-count version 0.11.0 with option -i set to "gene_id"[74]. The resulting count table was used for differential gene expression analysis with DESeq2 version 1.10.0 using Wald test[75]. Analysis and visualization of the differential expression data was performed with the R software package (version 2.15.1) using the cummeRbund library (version 2.0). Gene ontology was analyzed by GeneCoDis3, http://geneontology.org/ and wormbase web-based software. Statistical significance of the overlap between two groups of genes was calculated by a program available at: http://nemates.org/MA/progs/overlap_stats.html, assuming 20470 protein codding genes in *C. elegans*.

## Data availability

Source data are provided with this paper. RNA-seq data generated in this study and associated with Figs. 3, 4, 6 and 7, Supplementary Figs. 2, 3, 4, 5 and 6 have been deposited to NCBI with the accession code PRJNA714646.

## Code availability

Code for RNA-seq data processing and differential expression analysis is available on github (https://github.com/eco32i/cele-thiols.git).

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

## Acknowledgements

This work was supported by the Russian Science Foundation grants 19-74-30007 (A.A.M.) and 17-74-30030 (A.M., S.E., O.K.); Glenn Foundation for Medical Research, Blavatnik Family Foundation, and by the Howard Hughes Medical Institute (E.N.).

## Author contributions

I.G. and E.N. conceptualized this study. I.G. and E.N. designed the experiments. I.G., L.G., S.E., O.K., and A.M. performed all C. elegans experiments. I.S. performed sequencing and gene expression analysis. B.P. designed and performed the human tissue culture experiments. I.G., A.M., A.A.M., and E.N. analyzed data. I.G. and E.N. wrote the paper with input from all co-authors.

## Competing interests

The authors declare no competing interests.
