## [Peer Review File · Nature Communications]

Reviewer comments, first round

Reviewer #1 (Remarks to the Author):

In this manuscript, Gusarov and colleagues examine the effects of glutathione (GSH) and its precursor, N-acetyl cysteine (NAC) on the lifespan of *C. elegans*. Although this reviewer appreciates the efforts the authors have made to clearly delineate the effects of natural, bacteria-derived vs exogenously supplied thiols under different experimental conditions, I find the effects to be rather weak and unfortunately, the main claims of the paper to be not well supported by the data.

The authors show that supplementation of the plates with "fresh" NAC, as well as GSH, slightly shortens *C. elegans* lifespan under standard growth conditions. Feeding worms dead bacteria extends their lifespan, and under these conditions, NAC also shortens their lifespan. The degree of shortening is dependent on the concentration of NAC. (However, it is unclear to me why the authors did not also test whether the degree of shortening by NAC when the worms are fed on live bacteria is also concentration dependent.) From this, the authors hypothesise that when *C. elegans* are fed on live bacteria, their lifespan is limited by high levels of bacterially-derived glutathione.

To test this further, the authors quantify total and reduced glutathione in live vs dead bacteria, and *C. elegans* fed live vs dead bacteria. They claim that dead bacteria (as well as worms fed on dead bacteria) have lowered levels of glutathione and reduced thiols, and that the levels can be increased by supplementation with NAC. Although I agree that the data suggest this, the inherent noisiness in the measurements (and lack of statistical analysis) makes this not strongly convincing.

To further support the claim that bacterially-derived glutathione limits lifespan, the authors test the effects of the thiol-oxidising agent diamide on the lifespan of *C. elegans* fed live or dead bacteria. As has previously been shown, the authors find that diamide increases lifespan on live bacteria. However, they find it shortens lifespan on dead bacteria. It is unclear to me why that would be the case unless it is a case of both too much and too little being detrimental. A quantification with convincing statistical analysis of the effects of diamide on reduced thiols might help to clarify this point.

To work out the mechanism by which dietary thiols reduce lifespan, the authors use RNAseq to explore the effects of NAC supplementation on gene expression. The authors find that among the 1382 genes downregulated by NAC (when examining the effects of NAC supplementation on *C. elegans* fed dead bacteria), are genes that are known to be up-regulated by DAF-16 (180 genes) and SKN-1 (83 genes). However, NAC shortens the lifespan of *daf-16* even more strongly than wild-type, yet has no effect on *skn-1*. This analysis, which involves so very few genes out of the 1382, does not convince this reviewer that NAC shortens lifespan by the same mechanism as downregulation of DAF-16- and SKN-1-mediated transcriptional responses.

Even more strikingly, the authors claim that SKN-1 inhibition accounts for the accelerated aging caused by NAC. The additional support for this strong statement comes from the observation that there is a very modest overlap in genes that are up in mutants of WDR-23 (a regulator of SKN-1) compared to genes that are downregulated by NAC (229 genes out of the 1382 genes down in NAC-treated and over 2000 up in *wdr-23* mutants) and that the already shortened lifespan of *wdr-23*(RNAi) worms is not further shortened by NAC. Unfortunately, I do not find this very convincing. There is a lot of noise in transcriptomics and any very limited overlap cannot be rigorously interpreted to demonstrate any form causality due to changes in gene expression.

The authors go on to test the effects of inhibiting GSH import, using acivicin, which inhibits the

gamma-glutamyltransferase enzyme. The authors show that acivicin robustly increases the lifespan of *C. elegans* fed live bacteria. However, the mechanism for this is unclear. While acivicin suppresses the effect of GSH supplementation when *C. elegans* are fed dead bacteria, consistent with it interfering with GSH import, *C. elegans* fed live bacteria and treated with acivicin, have only a mild reduction in GSH levels at best. Given the strong effect of acivicin on worms fed live bacteria, it would certainly be interesting to explore this phenomenon further. It would also be reassuring to see that the effect is concentration dependent.

To elucidate the effect of GSH restriction on lifespan extension, the authors examine the gene expression changes brought about by acivicin. The authors find a very small number of gene expression changes and compare them to previously reported gene expression changes in *daf-2*, *clk-1*, *isp-1* and *nuo-6* mutants. Given the small numbers of genes involved and the fact that the authors themselves have shown that the growth conditions are important for the phenomenon they are studying, this analysis is again not very convincing. In particular, the finding that GO analysis of genes upregulated by acivicin suggests that the UPR is activated is by itself insufficient to conclude that the UPR is activated, or that UPR activation is extending lifespan here. While the authors find that the acivicin pre-treatment makes animals more resistant to select stressors, an involvement of the UPR would have to be examined rigorously to claim that this IS the mechanism by which GSH restriction mediates lifespan extension.

Reviewer #2 (Remarks to the Author):

This manuscript shows that dietary GSH shortens lifespan in *C. elegans*, while limiting GSH uptake extends lifespan, and increases stress resistance in mammalian cells. By examining transcriptome changes and performing genetic analyses, the authors conclude that the unfolded protein response and transcription factors SKN-1 and DAF-16 play key roles. The data provide a new twist on the familiar concept of hormesis, showing that well-studied protective pathways are influenced by the availability of GSH. While this is not necessarily surprising, the data provide a striking demonstration of why consumption of direct antioxidants is not beneficial, and can be deleterious, and may have influenced previous *C. elegans* aging studies. For these reasons the work is of substantial general interest and importance even though does not provides much in the way of new mechanistic insights.

Specific comments:

1. *C. elegans* lifespan can be extended by inhibition of germ cell proliferation. This can be seen even when genetic manipulations (or potentially drugs) are applied as late as the L4 stage (see original Kenyon lab papers). The authors need to provide controls that the various treatments used here aren't influencing lifespan by inhibiting reproduction.
2. I didn't see it explicitly stated whether FUdR was used in lifespan experiments. As the authors acknowledge, FUdR can have confounding effects. Key lifespans should be repeated without FUdR if this has not been done already.
3. A recent model posits that NAC effects involve its deacetylation to Cys, which can be converted to H₂S. Would this affect the authors' findings? Can this or conversion to GSH be accounted for?
4. I believe that there is a typo in Figure 1c. It should be 15mM NAC instead of NA.
5. The image and quantification graph in figures 5b and 5d do not match well, and no untreated is shown in b. Can the authors provide representative images for all conditions quantified in the graph? In the text it is stated that *tbb-1* expression changes 6-fold, but in the figure (5a) the difference is about 3-fold.
6. In their various quantification graphs in main and supplemental figures, have the authors conducted statistical analysis to show what changes are significant? Is there a particular reason that they only had P values in figures 4 and 6?
7. In figure 1g, what is the survival curve like under the condition of live OP50+NAC? Can stress resistance be further increased under such conditions compared to dead OP50+NAC or live OP50

without NAC?

8. The authors proposed that endogenous ROS levels are higher in worms fed with DB than those fed with live OP50 in the main text. In figure 7 legend, they also proposed that low GSH levels favor increased ROS production in mitochondria. Have they measured overall ROS levels or mitochondrial ROS production to confirm their hypothesis, either in worms or cells?

9. The authors showed that a subset of *skn-1* targets were down regulated by NAC treatment. Were those *skn-1* targets up regulated in the transcriptome after acivicin treatment? Considering the role for SKN-1 in the pathway proposed by the authors, is acivicin-induced longevity impaired in *skn-1* mutants?

10. In figures 6e and 6f, acivicin seems to decrease cell viability. Is that true? In figure 6f, did acivicin significantly improve cell survival upon heat shock by comparing the 2nd and 4th conditions?

11. Limiting GSH uptake increases resistance to heat in cells, while supplementing with NAC leads to increased oxidative stress resistance in worms (figure 1g). Does NAC supplementation affect heat stress resistance in worms as well? Do authors have any ideas why lower levels of thiols seem to have opposite effects on stress resistance in worms and cells? Comparing worms fed with dead OP50 and with or without NAC, how do they explain the fact that animals without NAC have increased sensitivity to paraquat but live longer than those with NAC (figure 1g)?

12. The authors found that acivicin only mildly decreased endogenous GSH levels in worms fed with LB (figure 4) FUDR and claimed that acivicin did not affect intracellular GSH levels in cells (without showing the data) in the Discussion. If endogenous GSH levels do not change, is GSH still important in the effects of acivicin on lifespan and stress resistance?

13. Have all major RNA-Seq results been verified by RT-PCR?

14. In their model (figure 7), the authors imply that when GSH levels are low, DAF-16 acts downstream of mitochondrial function and UPRER. Do they have evidence supporting that? Is the canonical UPRMT (ATFS-1) involved?

REVIEWER #1 (REMARKS TO THE AUTHOR):

In this manuscript, Gusarov and colleagues examine the effects of glutathione (GSH) and its precursor, N-acetyl cysteine (NAC) on the lifespan of *C. elegans*. Although this reviewer appreciates the efforts the authors have made to clearly delineate the effects of natural, bacteria-derived vs exogenously supplied thiols under different experimental conditions, I find the effects to be rather weak and unfortunately, the main claims of the paper to be not well supported by the data.

We appreciate the reviewer's thoughtful comments, which helped to improve our manuscript substantially. We have addressed all the reviewer's concerns as outlined below. In particular, we performed additional repeats and statistical analysis for *all* the experiments and also utilized an alternative method to directly measure thiols in *C. elegans*. All new results strongly support the claims we made in the original manuscript.

We respectfully disagree that the effects we report in the manuscript are "*rather weak*". It has been generally assumed that chronic supplementation with dietary antioxidants is beneficial for animal health. We show not only that the most popular thiol antioxidants failed to promote longevity of *C. elegans*, they consistently *reduced* the lifespan. We further demonstrate that the pharmacological *restriction* of dietary glutathione *increased* the lifespan by over 30%, which we believe is quite dramatic for the effect of a highly specific small molecule.

The authors show that supplementation of the plates with "fresh" NAC, as well as GSH, slightly shortens *C. elegans* lifespan under standard growth conditions. Feeding worms dead bacteria extends their lifespan, and under these conditions, NAC also shortens their lifespan. The degree of shortening is dependent on the concentration of NAC. (However, it is unclear to me why the authors did not also test whether the degree of shortening by NAC when the worms are fed on live bacteria is also concentration dependent.) From this, the authors hypothesise that when *C. elegans* are fed on live bacteria, their lifespan is limited by high levels of bacterially-derived glutathione.

As suggested by the reviewer, we now show the effect of 15 mM NAC on the aging of *C. elegans* fed on live bacteria (**new Fig. 1a**). The result demonstrates that the NAC dose dependently shortens the lifespan. Also, it argues that the negative effect of supplemented NAC adds to the effect of thiols consumed naturally from live bacteria.

To test this further, the authors quantify total and reduced glutathione in live vs dead bacteria, and *C. elegans* fed live vs dead bacteria. They claim that dead bacteria (as well as worms fed on dead bacteria) have lowered levels of glutathione and reduced thiols, and that the levels can be increased by supplementation with NAC. Although I agree that the data suggest this, the inherent noisiness in the measurements (and lack of statistical analysis) makes this not strongly convincing.

We are pleased that the reviewer appreciates our point and we understand the concerns regarding the noise in our data. Indeed, *C. elegans* are very heterogeneous in their response to stimuli (PMID: 16041374). To address the reviewer's point, we repeated the GSH and total thiol measurements. We ran the statistical significance tests for all the experiments, determined *p*-values, and presented our results in the graphs as mean \pm standard error of mean (**revised Fig. 2a and b**). The data on individual repeats are now presented in **new Supplementary Table 6**. These results clearly support our original claim that worms fed on a dead bacteria diet accumulate significantly less reduced thiols and less GSH (**revised Fig. 2b**), whereas NAC supplementation increased both cellular thiols and GSH.

To further validate our results, we used another independent method - the fluorescent dye-based assay. Worms fed various diets were treated with ThioFluor 623 (Cayman Chemical), which

specifically interacts with reduced thiols (**new Fig. 2d and e**). These results are highly consistent with our biochemical thiol measurements (**Fig. 2b, d and e**) and demonstrate that worms reared on dead bacteria accumulate less thiols. The raw data on thiol staining are now presented in **new Supplementary Table 7**.

To further support the claim that bacterially-derived glutathione limits lifespan, the authors test the effects of the thiol-oxidising agent diamide on the lifespan of *C. elegans* fed live or dead bacteria. As has previously been shown, the authors find that diamide increases lifespan on live bacteria. However, they find it shortens lifespan on dead bacteria. It is unclear to me why that would be the case unless it is a case of both too much and too little being detrimental. A quantification with convincing statistical analysis of the effects of diamide on reduced thiols might help to clarify this point.

The reviewer is correct about the nonlinear effect of thiol oxidation on the lifespan. We have shown previously that while the low level of diamide (5 mM) extended the lifespan, the higher level (15 mM) shortened it (PMID: 28627510). Congruently, Urban et al. demonstrated that another thiol depleting chemical, diethyl maleate, extends the lifespan at low (0.1 mM) concentration, while accelerating the aging at higher doses (1mM) (PMID: 28086197). Together, these data indicate that a slight decrease of endogenous thiols is beneficial, while severe depletion is detrimental (This is not surprising considering the critical role of endogenous thiols in various metabolic processes). In support of these observations, we found that 5 mM diamide and dead *E. coli* OP50 diet reduce the thiol level in *C. elegans* to approximately the same extent (**new Fig. 2d and e**). On the contrary, 15 mM diamide almost completely eliminate thiol staining by ThioFluor 623 (**Fig. 2d and e**).

To work out the mechanism by which dietary thiols reduce lifespan, the authors use RNAseq to explore the effects of NAC supplementation on gene expression. The authors find that among the 1382 genes downregulated by NAC (when examining the effects of NAC supplementation on *C. elegans* fed dead bacteria), are genes that are known to be up-regulated by DAF-16 (180 genes) and SKN-1 (83 genes). However, NAC shortens the lifespan of *daf-16* even more strongly than wild-type, yet has no effect on *skn-1*. This analysis, which involves so very few genes out of the 1382, does not convince this reviewer that NAC shortens lifespan by the same mechanism as downregulation of DAF-16- and SKN-1-mediated transcriptional responses.

Even more strikingly, the authors claim that SKN-1 inhibition accounts for the accelerated aging caused by NAC. The additional support for this strong statement comes from the observation that that there is a very modest overlap in genes that are up in mutants of WDR-23 (a regulator of SKN-1) compared to genes that are downregulated by NAC (229 genes out of the 1382 genes down in NAC-treated and over 2000 up in *wdr-23* mutants) and that the already shortened lifespan of *wdr-23*(RNAi) worms is not further shortened by NAC. Unfortunately, I do not find this very convincing. There is a lot of noise in transcriptomics and any very limited overlap cannot be rigorously interpreted to demonstrate any form causality due to changes in gene expression.

It is not surprising that after 8 days of exposure to NAC worms adjust their global gene expression (1383 genes are downregulated). However, it is highly unlikely that most of these genes affect the worm's lifespan. We found the subsets of *daf-16* and *skn-1*-dependent genes among the genes downregulated by NAC. In the revised manuscript we used the resource available online to calculate the statistical significance of the overlap between the two groups of genes (http://nemates.org/MA/progs/overlap_stats.cgi). The results indicate that 92 gene overlap (between 1383 downregulated by NAC and 232 upregulated by SKN-1, assuming the total of 20470 protein coding genes in *C. elegans* genome) is 5.9 times larger than an overlap expected by chance between these two groups, and is highly statistically significant (Representation factor: 5.9 p -value < 3.012e-47). The

overlap between DAF-16 upregulated and NAC downregulated genes was also statistically significant (Representation factor: 5.6, p -value < 4.277e-93). This analysis is now presented in **revised Fig. 3 legend**.

To further estimate the noise in our RNA-seq data we used a more stringent cut off value of $q < 0.0001$ (for genes downregulated more than two folds). That decreased the number of genes downregulated by NAC from 1383 to 1047, i.e. by 24.3%. We should expect the same percentage decrease in the overlap between the genes downregulated by NAC and upregulated by SKN-1, if this overlap were random. However, the new overlap between NAC downregulated 1047 genes by SKN-1 upregulated 232 genes decreased from 92 to 79, i.e. by 14.1%. Statistical analysis demonstrates that this overlap is still highly significant ($p < 7.455e-44$), and the representation factor is even larger (6.7). This exercise demonstrates that the cut-off by the $q < 0.05$ provides an optimal balancing between the noise reduction and compromising the results.

We agree with the reviewer that differential expression analysis is indicative and must be experimentally verified. As the activity of both *daf-16* and *skn-1* determines *C. elegans* lifespan, we tested whether worms depleted of these factors would be sensitive to NAC. We found that the lifespan of *daf-16* worms can be further shortened by NAC (**Fig. 3d**). In contrast, the lifespan of *skn-1* worms was not shortened by NAC (**Fig. 4d**). Moreover, the expression of *gst-4* (marker of SKN-1 activity) was repressed by both bacterial and supplemental thiols (**revised Fig. 4a and b**). This led us to conclude that the repression of SKN-1 activity by NAC shortens the *C. elegans* lifespan. Together, these experiments allowed us to verify the leads we obtained from the differential expression analysis.

The authors go on to test the effects of inhibiting GSH import, using acivicin, which inhibits the gamma-glutamyltransferase enzyme. The authors show that acivicin robustly increases the lifespan of *C. elegans* fed live bacteria. However, the mechanism for this is unclear. While acivicin suppresses the effect of GSH supplementation when *C. elegans* are fed dead bacteria, consistent with it interfering with GSH import, *C. elegans* fed live bacteria and treated with acivicin, have only a mild reduction in GSH levels at best. Given the strong effect of acivicin on worms fed live bacteria, it would certainly be interesting to explore this phenomenon further. It would also be reassuring to see that the effect is concentration dependent.

As suggested by reviewer, we treated worms with escalating concentrations of acivicin and stained them for endogenous thiols (**new Fig. 5a**). As expected, the amount of thiols in *C. elegans* decreased proportionally to the concentration of acivicin.

GSH concentration was reduced only slightly in acivicin-treated worms (**revised Extended Data Fig. 7b**), indicating that GSH is synthesized endogenously, most probably at the expense of other Cys consuming processes. This result is consistent with previously published data that acivicin did not decrease GSH concentration, but decreased Cys concentration in human cells (PMID: 15469854).

To elucidate the effect of GSH restriction on lifespan extension, the authors examine the gene expression changes brought about by acivicin. The authors find a very small number of gene expression changes and compare them to previously reported gene expression changes in *daf-2*, *clk-1*, *isp-1* and *nuo-6* mutants. Given the small numbers of genes involved and the fact that the authors themselves have shown that the growth conditions are important for the phenomenon they are studying, this analysis is again not very convincing.

We included a statistical analysis in the revised manuscript demonstrating that the overlap between the genes upregulated by acivicin and by the mitochondrial mutations is more than random and statistically significant (see **revised Fig. 6 and its legend**). We agree that this overlap alone may not be sufficient to claim that the GSH restriction affects mitochondria. However, a combination of several of our experiments supports this conclusion: 1) In the revised manuscript we present evidence that acivicin induces mitochondrial ROS production, which has been associated with longer lifespan (**new Fig. 6d**); 2)

Acivicin induces *tbb-6* expression, which is also upregulated by many mutations and RNAi that cause mitochondrial disturbances (PMID: 27420916); 3) The mitochondrial uncoupler, DNP, induces *tbb-6* expression, thus providing additional evidence that *tbb-6* is a marker of mitochondrial disturbance; 4) A dead bacteria diet (which is low on thiols) does not increase the lifespan of *nuo-6* mutant worms (**Extended Data Fig. 8b**), while NAC shortens *nuo-6* lifespan (PMID: 21151885).

In particular, the finding that GO analysis of genes upregulated by acivicin suggests that the UPR is activated is by itself insufficient to conclude that the UPR is activated, or that UPR activation is extending lifespan here. While the authors find that the acivicin pre-treatment makes animals more resistant to select stressors, an involvement of the UPR would have to be examined rigorously to claim that this IS the mechanism by which GSH restriction mediates lifespan extension.

The resistance to proteotoxic stresses correlates well with the longer lifespan. Usually the activation of HSF1 increases stress resistance and promotes longevity. However, our results in *C. elegans* and human cells indicate that acivicin promotes an HSF1-independent response. Typical HSF1-dependent markers of ER- and Mito-UPR, *hsp-4*, and *hsp-6* were not upregulated by acivicin (**Supplementary Table 4 and new Extended Data Fig. 9**). In human cells, the major regulator of ER-UPR, XBP1, and some typical ER chaperones were upregulated by acivicin (**Supplementary Table 4**). However, the knock down of HSF1 did not compromise acivicin induced heat stress resistance (**new Extended Data Fig. 11**). These results allowed us to suggest in the discussion that the acivicin effects on stress resistance and longevity are in the same pathway as *daf-2*. It has been shown that in long-lived *daf-2* worms, DAF-16 and XBP-1 function together to promote proteotoxic stress resistance and longevity without upregulating ER HSPs (PMID: 20460307).

We agree that the exact mechanism of UPR induction by acivicin and its impact on life extension have not been completely elucidated in this manuscript. It would be interesting to uncover the mechanistic details of the effect of acivicin on XBP-1 and DAF-16. However, in the view of potential complexity of this mechanism, we believe these studies are well beyond the scope of this manuscript.

REVIEWER #2 (REMARKS TO THE AUTHOR):

This manuscript shows that dietary GSH shortens lifespan in *C. elegans*, while limiting GSH uptake extends lifespan, and increases stress resistance in mammalian cells. By examining transcriptome changes and performing genetic analyses, the authors conclude that the unfolded protein response and transcription factors SKN-1 and DAF-16 play key roles. The data provide a new twist on the familiar concept of hormesis, showing that well-studied protective pathways are influenced by the availability of GSH. While this is not necessarily surprising, the data provide a striking demonstration of why consumption of direct antioxidants is not beneficial, and can be deleterious, and may have influenced previous *C. elegans* aging studies. For these reasons the work is of substantial general interest and importance even though does not provides much in the way of new mechanistic insights.

We appreciate the reviewers' enthusiastic evaluation of our work and detailed and constructive comments. We tried our best to address all of them in full in the revised manuscript.

Specific comments:

1. *C. elegans* lifespan can be extended by inhibition of germ cell proliferation. This can be seen even when genetic manipulations (or potentially drugs) are applied as late as the L4 stage (see original

Kenyon lab papers). The authors need to provide controls that the various treatments used here aren't influencing lifespan by inhibiting reproduction.

Both DAF-16 and SKN-1 transcription factors are required to extend the lifespan of germless worms (PMID: 16530050, PMID: 27140632). If acivicin extended the lifespan by suppressing germline signaling, we would expect both of these transcription factors to limit its antiaging effect. We showed that acivicin does not extend the lifespan of *daf-16* worms (Fig. 5e). However, in the revised manuscript we demonstrate that acivicin does increase the lifespan of the *skn-1* mutant (new Fig. 5d). This was consistent with our transcriptomic results demonstrating that acivicin did not induce major SKN-1 target genes (Fig. 6a) or *hsp-6* (new Extended Data Fig. 9c, d), which are all upregulated by germ cell loss.

Acivicin slightly decreased the lifespan of *glp-1* worms (new Extended Data Fig. 10b). Moreover, we found that *glp-1* animals have a very low level of reduced thiols (new Extended Data Fig. 10c, d). A small increase in ROS production is usually beneficial, but the higher ROS concentration can be detrimental. As both acivicin and the *glp-1* mutation decrease thiols and increase ROS (new Fig. 6d and Extended Data Fig. 10c, d), we hypothesize that such double treatment elevates ROS beyond its beneficial level. Similar interaction has been found between *glp-1* and paraquat (PQ): Individually, both PQ and *glp-1* increased ROS and extended the lifespan (PMID: 27140632), however, their combination did not (PMID: 27140632). Taken together our results suggest that acivicin does not inhibit germ cell signaling, but affects the same downstream targets (via ROS and DAF-16).

As it has been shown that NAC supplementation suppressed anti-aging ROS signaling to shorten the long life of germless (*glp-1*) worms (PMID: 27140632), we believe NAC does not act through the inhibition of reproduction.

2. I didn't see it explicitly stated whether FUdR was used in lifespan experiments. As the authors acknowledge, FUdR can have confounding effects. Key lifespans should be repeated without FUdR if this has not been done already.

All lifespan experiments were carried out without FUdR, except for the following two:

1. We used 40 μ M FUdR for the experiment shown in Fig. 1c. As bacteria were heat inactivated for this experiment we used FUdR to prevent excessive internal hatching. This low concentration of FUdR does not affect the lifespan and does not reverse the NAC effect.
2. We used 100 μ M FUdR in the experiment shown in Extended Data Fig 1b to demonstrate that at this level FUdR extends the lifespan and has an adverse interaction with NAC.

3. A recent model posits that NAC effects involve its deacetylation to Cys, which can be converted to H₂S. Would this affect the authors' findings? Can this or conversion to GSH be accounted for?

The reviewer is correct. Judging by its characteristic smell, some NAC must be converted to H₂S by bacteria. It has been demonstrated that H₂S extends the *C. elegans* lifespan (PMID: 18077331), although we show here that NAC shortens it. Thus, we would expect that the negative effect of NAC can be even more damaging if H₂S were not produced. This, actually, may explain why 15 mM NAC decreases the lifespan of *C. elegans* fed on dead bacteria more than it does when *C. elegans* is fed on live bacteria (38% vs 27%, compare revised Fig. 1a and c).

A recent study demonstrated that CBS-1 is capable of H₂S production from Cys in *C. elegans* crude extracts *in vitro* (PMID: 27140632). However, the main function of this enzyme *in vivo* is to produce cystathionine from homocysteine (no H₂S is produced in this reaction). This important reaction controls the level of homocysteine and may be a factor that affected the lifespan independently of H₂S. However, another study claims that MPST-1 is a key H₂S producing enzyme that affects the lifespan (PMID: 24093496). From our own extensive experience in the study of H₂S, we know that its biogenesis

is extremely complicated (at least 3 independent enzymes can synthesize it along with many less characterized desulfurases) and interconnected with Cys, Met, SAM and THF metabolism and methylation. Changes in the concentration of any of these metabolites can affect the lifespan independently of H₂S. It will take many months or more to untangle these interactions. We believe that the studies of the role of H₂S in NAC-mediated lifespan shortening are beyond the scope of this manuscript.

4. I believe that there is a typo in Figure 1c. It should be 15mM NAC instead of NA.

Thank you for identifying our mistake. We corrected it.

5. The image and quantification graph in figures 5b and 5d do not match well, and no untreated is shown in b. Can the authors provide representative images for all conditions quantified in the graph? In the text it is stated that tbb-1 expression changes 6-fold, but in the figure (5a) the difference is about 3-fold.

We apologize for making the image labeling difficult to read. We tried our best to improve it (**revised Fig. 6b**). The image shows the untreated worms – it is the second group from the left (**Fig. 6b**). We did not show the image for 5 mM NAC treatment only. There are no differences between untreated and 15 mM NAC treated worms and, thus, it would be redundant. We now provide raw GFP intensities and statistical data for Figs. 6b-d in **the Supplementary Tables 11 and 12**.

The fold changes in **Fig. 6a** are in log₂ scale ($2^{2.84}=7.16$ fold change). We now indicate this in the **Fig. 6a** legend.

6. In their various quantification graphs in main and supplemental figures, have the authors conducted statistical analysis to show what changes are significant? Is there a particular reason that they only had P values in figures 4 and 6?

In the revised manuscript, we now show *p*-values for every graph and provide the raw data in **Extended Tables 6-15**.

7. In figure 1g, what is the survival curve like under the condition of live OP50+NAC? Can stress resistance be further increased under such conditions compared to dead OP50+NAC or live OP50 without NAC?

Yes, in the revised manuscript, we demonstrate that NAC supplementation can further increase oxidative stress resistance on a live *E. coli* diet (**new Extended Data Fig. 1c**). The effect is not as dramatic as on a dead bacteria, as live bacteria provide worms with GSH. Thus, the accumulation of higher endogenous thiols either from the bacterial diet or from a combination of bacterial and supplemented thiols provide relief from acute oxidative stress, but is detrimental in a long term (interfere with antioxidant gene expression (**Supplementary Table 1**), and shortens the lifespan (**Fig. 1a-c**)).

8. The authors proposed that endogenous ROS levels are higher in worms fed with DB than those fed with live OP50 in the main text. In figure 7 legend, they also proposed that low GSH levels favor increased ROS production in mitochondria. Have they measured overall ROS levels or mitochondrial ROS production to confirm their hypothesis, either in worms or cells?

We agree that this is a very important point. In the revised manuscript, we demonstrate that feeding worms dead *E. coli* increases ROS, while NAC suppresses ROS (**new Fig. 4e,f**). Moreover, acivicin treatment depletes thiols (**new Fig. 5a**) and increases mitochondrial ROS (**new Fig. 6d**).

9. The authors showed that a subset of *skn-1* targets were down regulated by NAC treatment. Were those *skn-1* targets up regulated in the transcriptome after acivicin treatment? Considering the role for SKN-1 in the pathway proposed by the authors, is acivicin-induced longevity impaired in *skn-1* mutants?

Although 12 out of 34 acivicin induced genes were downregulated by NAC, the SKN-1 targets were not enriched among the genes upregulated by acivicin. Accordingly, we now demonstrate that acivicin increases the lifespan of *skn-1* mutant worms (**new Fig. 5d**). Taken together, our results demonstrate that high endogenous thiols suppress SKN-1, while thiol restriction promotes DAF-16 activity.

10. In figures 6e and 6f, acivicin seems to decrease cell viability. Is that true? In figure 6f, did acivicin significantly improve cell survival upon heat shock by comparing the 2nd and 4th conditions?

Indeed, acivicin treatment decreases cell count. To investigate this, we first monitored the cell growth and apoptosis in the presence of an escalating amount of acivicin. It appears that acivicin inhibits cell proliferation (**new Extended Data Fig. 11a**) without significant cell death (**new Extended Data Fig. 11b**). As some apoptosis does occur at the highest acivicin dose (100 μ M) and longer incubation time, we decided to repeat the stress resistance assay after pretreatment with lower doses of acivicin (10 and 60 μ M) for 24 hours. As shown in **revised Fig. 7f,g** and **new Extended Data Fig. 11c,d**, acivicin pretreatment promotes significant increase in cell survival.

To address the second part of the question, there is a statistically significant increase in the number of live cells after heat shock between control (432 \pm 44cells) and 10 μ M (685 \pm 21, p-value=0.0068) or 60 μ M (642 \pm 25, p-value=0.014) acivicin pretreated samples (conditions 2, 4 and 6 in **new Extended Data Fig. 11d**). However, we believe the percent survival better represents our results (**new Fig. 7f,g**), as it accounts for the original number of cells before stress.

11. Limiting GSH uptake increases resistance to heat in cells, while supplementing with NAC leads to increased oxidative stress resistance in worms (figure 1g). Does NAC supplementation affect heat stress resistance in worms as well? Do authors have any ideas why lower levels of thiols seem to have opposite effects on stress resistance in worms and cells? Comparing worms fed with dead OP50 and with or without NAC, how do they explain the fact that animals without NAC have increased sensitivity to paraquat but live longer than those with NAC (figure 1g)?

Organisms employ different strategies to cope with stress. For example, cells can be similarly protected against heat shock either by high intracellular glycerol or by the induction of chaperones. While having extra chaperones would be beneficial for a longer life, having extra glycerol for a long period of time would be detrimental, as it feeds into the carbohydrate metabolic pathways. We propose that a similar effect occurs in thiol excess. While accumulated extra thiols protect against acute insult (paraquat), they chronically scavenge beneficial ROS, which regulate anti-aging signaling.

In the revised manuscript, we demonstrate that worms reared on dead bacteria have less endogenous thiols and are more resistant to heat stress (**new Fig. 7e**). Furthermore, adding NAC sensitizes them to heat stress (**Fig. 7e**). The mechanisms of oxidative and heat stress resistance are very different and independent. Moreover, it has been suggested that there is a strong correlation between heat-, but not oxidative stress-, resistance and longer lifespan (PMID: 31309734).

Accordingly, we show that thiol restriction increases heat- and tunicamycin-resistance in both worms and human cells (**new Fig. 7c-g**).

12. The authors found that acivicin only mildly decreased endogenous GSH levels in worms fed with LB (figure 4) FUdR and claimed that acivicin did not affect intracellular GSH levels in cells (without showing

the data) in the Discussion. If endogenous GSH levels do not change, is GSH still important in the effects of acivicin on lifespan and stress resistance?

We now demonstrate that an acivicin dose dependently decrease the level of reduced thiols in *C. elegans* (**new Fig. 5a**). However, the GSH level did not decrease as much (**Extended Data Fig. 7b**). These results are consistent with previously published data that acivicin decreases the cellular Cys level without significantly affecting intracellular GSH (PMID: 15469854), indicating that Cys use is prioritized for GSH synthesis. We also found that the inhibition of glutathione biosynthesis by *gcs-1* RNAi did not upregulate *tbb-6::GFP* expression (**new Extended Data Fig. 7c**). Together, these data propose that the overall level of reduced endogenous thiols is more important for the lifespan than is only GSH.

13. Have all major RNA-Seq results been verified by RT-PCR?

We used fluorescent GFP promoter fusion strains to verify the key results of RNA-seq (*gst-4* and *tbb-6*). We used 3 to 6 biological repeats and stringent statistical criteria in our transcriptomic experiments, which make these results more reliable compared to RT-PCR, especially for those genes whose expression changes modestly. In RNA-seq all reads aligned to the sequence of a gene of interest are counted and normalized to an internal control, such as the total number of reads. This makes the results precise and specific. On the contrary, RT-PCR does not have an internal standard (expression of control genes might be affected by treatment) and is progressively noisy at lower levels of expression.

14. In their model (figure 7), the authors imply that when GSH levels are low, DAF-16 acts downstream of mitochondrial function and UPRER. Do they have evidence supporting that? Is the canonical UPRMT (ATFS-1) involved?

Our experiments demonstrate that *daf-16* is required for longer lifespan of GSH restricted worms (**Fig. 5e**). Also GSH restriction induces non-canonical mitochondrial and ER stress responses. Chaperones typical for ER-UPR and Mit-UPR were not induced by acivicin (**Fig. 6a, new Extended Data Fig. 9**). However worms were more resistant to tunicamycin and long lived (**Figs. 5 and 7**). Previously, it was demonstrated that *daf-2* worms are tunicamycin resistant, long lived and do not express typical chaperones (*hsp-4*) (PMID: 20460307). That allowed us to hypothesize that the mechanism might be at least partially similar between GSH restriction and *daf-2*. Of course, it would be very interesting to investigate this further.

The upregulation of *tbb-6* also indicates that GSH restriction induces non canonical mitochondrial stress response. This gene is ATFS-1 independent and regulated by 38 MAP kinase PMK-3 (PMID: 27420916).

Reviewer comments, second round

Reviewer #1 (Remarks to the Author):

In their revised manuscript, Gusarov and colleagues have added several experiments which help to strengthen the central claim of their paper: that bacterially-derived glutathione limits lifespan in *C. elegans*.

The authors have also added experimental data to work out the mechanism by which acivicin, which inhibits GSH import by inhibiting the gamma-glutamyltransferase enzyme, extends lifespan. Using the dye ThioFluor 623, the authors demonstrate that acivicin decreases the total level of reduced thiols in live animals. It also increases staining by MitoTracker CM-H2X and the expression of a *tbb::gfp* reporter, which they show is also upregulated by acetaminophen and cadmium, which are known to deplete GSH.

The authors have also added statistical analyses throughout and included raw data in supplementary tables. Despite the added statistics, I still remain unconvinced that the RNAseq data allows the authors to reach strong conclusions about the underlying mechanisms. For example, the strong claim that "SKN-1 inhibition accounts for the accelerated aging caused by NAC" comes from the observation that 40% of genes that are upregulated by SKN-1 are downregulated by NAC, that there is a very modest overlap in genes that are up in mutants of WDR-23 (a regulator of SKN-1) compared to genes that are downregulated by NAC (229 genes out of the 1382 genes down in NAC-treated and over 2000 up in *wdr-23* mutants), and that the already shortened lifespan of *wdr-23(RNAi)* worms is not further shortened by NAC. Given how many genes are affected by NAC, it seems unlikely that SKN-1 inhibition is the only target. For example, the authors show there is also overlap between DAF-16 targets and NAC. Thus, there are likely to be many targets.

While the authors find a robust effect of acivicin on lifespan extension, they find only a very modest number of genes changed yet attempt to work out its mechanism of action by looking at overlaps between the set of 39 genes changed in acivicin and the hundreds to thousands of genes changed by mitochondrial mutants and *daf-2* mutants. Given the small number of genes changed, one almost wonders if the acivicin failed to work in the RNAseq experiment and if there is any meaning at all in this analysis. (I do however acknowledge that *tbb-6* was one of the genes found in the RNAseq analysis.). I find the *in vivo* data for acivicin is more convincing. Yet the array of experiments the authors perform in an attempt to link acivicin with many pathways (mitochondrial signalling, mtUPR, ER stress response) feels a bit *ad hoc*. Moreover, given that acivicin doesn't have its full effect on either DAF-16 or SKN-1 (and as discussed above there is overlap between DAF-16 and SKN-1 targets and gene expression changes brought about by NAC treatment) it seems artificial to place SKN-1 in the high thiol pathway and DAF-16 in the low thiol pathway.

Overall, all the analyses pertaining to RNAseq data feel stretched and unconvincing.

In sum, I believe the paper presents interesting findings that could be of interest to the worm aging and ROS fields. However, I also strongly believe the authors should tone down the claims about the mechanisms involved, including a more reasonable interpretation of the effects of acivicin.

Reviewer #2 (Remarks to the Author):

The paper is much improved and will be a provocative and important study because it addresses directly how antioxidant administration affects life/healthspan and cellular homeostasis mechanisms. Although much remains to be learned, this work establishes that this will be significant area to pursue. I have just a few remaining textual concerns that should still be

addressed.

Remaining concerns:

1. Previous concern 1: It may just be the way the relevant sentence is written, but the authors appear to claim in the text that acivicin "slightly" decreased the number of progeny. However, extended figure 10a shows that acivicin decreases brood size to about 28%, and that the difference is significant. They should insert the word "significantly" after "progeny" in the relevant sentence. In any event, their strongest argument that their lifespan extension does not involve reproductive suppression is its independence from *skn-1*, which does suffice.
2. Previous concern 2: the authors in their response claim that 40 μ M FuDR does not affect lifespan and does not reverse the NAC effect. Could the authors provide or point out the data supporting this claim? This is important because they used 40 μ M FuDR to measure the effect of NAC on the transcriptome.
3. Previous concern 10: Could the authors clarify how they calculated cell survival in figures 7f and 7g? Did they use the number of cells after acivicin treatment but before stress, or the number of cells they plated? 24 hrs is long enough for most cells to go through a cell cycle, and cell numbers double (extended figure 11a). Besides, 24hr acivicin treatment seems to inhibit cell proliferation and alters the number of cells (extended figure 11a). Why did the authors use such a long treatment? Is it because shorter treatments do not affect GSH uptake? Is it established that the condition they used decreases intracellular GSH or reduced thiol levels? Did they also use the same condition for RNA-Seq? Survival is misspelled in panel f.
4. It has been shown that moderate SKN-1 overexpression increases lifespan (Tullet et al, Cell, 2008), and the Choe lab reported previously that *wdr-23* mutation or knockdown increased lifespan and GSH levels in a *skn-1*-dependent manner (PMID 26056713). Do the authors have thoughts on why *wdr-23* KD shortened lifespan here (supplementary table 5)? The authors indicate that NAC suppresses *skn-1* to decrease lifespan. Why did NAC not shorten the lifespan of *wdr-23* KD worms? Also, why was this lifespan experiment performed at 25C instead of their typical temperature of 20C?
5. In the last sentence of page 11, the statement that "the downregulation of *daf-16*- and *skn-1*-mediated response shortens the lifespan of animals fed LB by the same mechanism as does NAC" is not very convincing. First, NAC still shortens the lifespan of *daf-16* mutant worms (figure 3). Second, a large number of genes changed by NAC in the transcriptome are not DAF-16 or SKN-1 targets (figure 3c). I agree that *skn-1* suppression may mediate the effect of NAC to some extent, but their statement is too strong.
6. The authors establish that high levels of antioxidant thiols reduce lifespan by inhibiting SKN-1 (NRF2), but the conventional thinking would be that SKN-1 otherwise increases lifespan by bolstering antioxidant defenses. This suggests that other functions of SKN-1 might be the most pertinent for aging/longevity. It would be good for the authors to point this out more explicitly and if they like speculate on what these functions might be.

Minor issues:

1. Previous concern 5: In the response, the authors state that 15 mM NAC worms show no difference in *tbb-6::GFP* intensity compared to untreated worms. This is supported by the image in figure 6b and the quantification graph. However, in the manuscript (on page 14) the authors claim that NAC decreased *tbb-6::GFP* fluorescence. They should clarify if they intend this to apply only to acivicin-treatment conditions.
2. Can the authors clarify the statement "acivicin slightly delayed development" in more detail? This is relevant to the increased stress resistance of acivicin-treated worms than untreated worms. This is also relevant to the lifespan assay because acivicin treatment increased lifespan about 2 days when started at eggs than L4 (Extended table 5).
3. In the abstract, the statement that "the long-lasting effect of GSH or NAC on healthy animals has not been investigated" is not accurate. It has been reported that NAC has no effect on the lifespan of wild-type worms in the presence of FuDR (Wei and Kenyon, PNAS, 2016). Also, as the authors mentioned, NAC abolishes the beneficial effects of exercises (their reference 4). I recommend "the long-lasting effect of GSH or NAC on healthy animals has not been intensively investigated".
4. The sentence in the second and third lines of page 6 that "Thiol level in worms reared on LB+5 mM diamide was very similar to that in worms reared on DB (Fig 2d and e), indicating that the further depletion of thiols is not beneficial" needs to be explained in more detail.

5. For lifespan results in figure 5b-e, extended figure 5c-d, are the red numbers in the graphs changes in % or absolute numbers of days?

6. The statement on page 15 that "Accordingly, feeding worms with DB - a diet deficient in low molecular weight (LMW) thiols, markedly increases the lifespan of wt, but not nuo-6 worms (Extended Data Fig. 8a and b)" is not clear. According to the extended data Fig. 8, DB still increased the lifespan of nuo-6 worms, albeit to a lesser extent.

7. In the first line of page 18, the authors need to clarify what is GGT1.

8. In addition, why did the authors use one-tailed t-test in their lifespan statistical analysis? One-tailed t-test hypothesizes that the change will only go to one direction, while two-tailed t-test is more proper here because the investigators would not know in which direction the change might go (whether lifespan will increase or decrease).

Reviewer #1 (Remarks to the Author):

In their revised manuscript, Gusarov and colleagues have added several experiments which help to strengthen the central claim of their paper: that bacterially-derived glutathione limits lifespan in *C. elegans*.

The authors have also added experimental data to work out the mechanism by which acivicin, which inhibits GSH import by inhibiting the gamma-glutamyltransferase enzyme, extends lifespan. Using the dye ThioFluor 623, the authors demonstrate that acivicin decreases the total level of reduced thiols in live animals. It also increases staining by MitoTracker CM-H2X and the expression of a *tbb::gfp* reporter, which they show is also upregulated by acetaminophen and cadmium, which are known to deplete GSH.

The authors have also added statistical analyses throughout and included raw data in supplementary tables. Despite the added statistics, I still remain unconvinced that the RNAseq data allows the authors to reach strong conclusions about the underlying mechanisms. For example, the strong claim that “SKN-1 inhibition accounts for the accelerated aging caused by NAC” comes from the observation that 40% of genes that are upregulated by SKN-1 are downregulated by NAC, that there is a very modest overlap in genes that are up in mutants of WDR-23 (a regulator of SKN-1) compared to genes that are downregulated by NAC (229 genes out of the 1382 genes down in NAC-treated and over 2000 up in *wdr-23* mutants), and that the already shortened lifespan of *wdr-23*(RNAi) worms is not further shortened by NAC. Given how many genes are affected by NAC, it seems unlikely that SKN-1 inhibition is the only target. For example, the authors show there is also overlap between DAF-16 targets and NAC.

Thus, there are likely to be many targets.

The strongest indication that the inhibition of SKN-1 by NAC accounts for the shortened lifespan is provided by the experiments of Fig 4b-d, demonstrating that NAC inhibits the expression of *gst-4* and does not shorten the lifespan of *skn-1* worms. We agree with the reviewer that judging by a massive change in global transcription there must be other targets/pathways regulated by NAC. For example, the stronger negative effect of NAC on *daf-16* worms indicates that DAF-16 is involved. We amended the text to emphasize that SKN-1 is not the only NAC target.

While the authors find a robust effect of acivicin on lifespan extension, they find only a very modest number of genes changed yet attempt to work out its mechanism of action by looking at overlaps between the set of 39 genes changed in acivicin and the hundreds to thousands of genes changed by mitochondrial mutants and *daf-2* mutants. Given the small number of genes changed, one almost wonders if the acivicin failed to work in the RNAseq experiment and if there is any meaning at all in this analysis. (I do however acknowledge that *tbb-6* was one of the genes found in the RNAseq analysis.). I find the *in vivo* data for acivicin is more convincing. Yet the array of experiments the authors perform in an attempt to link acivicin with many pathways (mitochondrial signalling, mtUPR, ER stress response) feels a bit *ad hoc*. Moreover, given that acivicin doesn't have its full effect on either DAF-16 or SKN-1 (and as discussed above there is overlap between DAF-16 and SKN-1 targets and gene expression changes brought about by NAC treatment) it seems artificial to place SKN-1 in the high thiol pathway and DAF-16 in the low thiol pathway.

Indeed, acivicin affects a relatively small number of genes. Note, however, that *daf-2* and mitochondrial mutations increase the lifespan much stronger, which correlates with many more genes to be affected. Moreover, a significant portion of transcriptional changes associated with mitochondrial mutants must be to compensate for altered oxidative phosphorylation. As a process of adaptation to genetic or environmental perturbation is complex, it is hard to estimate a number of genes that have to be induced/repressed to prolong the lifespan for a certain amount of time.

We would like to emphasize that most of the acivicin-mediated life extension was lost in *daf-16* worms (31.7% increase in N2 vs 6.1% in *daf-16*). This result clearly demonstrates that DAF-16 is required for the anti-aging effect of acivicin. However, acivicin extends the lifespan of *skn-1* worms almost as robustly as that of the wt (15.7% in *skn-1* vs 18.2% in N2). It is likely that both *daf-16* and *skn-1* are regulated by low/high thiols pathways. Not all of this regulation is translated to, or meant to control, the lifespan.

Overall, all the analyses pertaining to RNAseq data feel stretched and unconvincing.

Our RNA-seq raw data will be available to the community upon the publication. We welcome the future efforts by expert bioinformaticians to gain more insight into the lifespan regulation by thiols.

In sum, I believe the paper presents interesting findings that could be of interest to the worm aging and ROS fields. However, I also strongly believe the authors should tone down the claims about the mechanisms involved, including a more reasonable interpretation of the effects of acivicin.

We thank the reviewer for their thoughtful and positive evaluation of the revised manuscript. We have modified the text to accommodate the reviewer's requests and suggestions.

Reviewer #2 (Remarks to the Author):

The paper is much improved and will be a provocative and important study because it addresses directly how antioxidant administration affects life/healthspan and cellular homeostasis mechanisms. Although much remains to be learned, this work establishes that this will be significant area to pursue. I have just a few remaining textual concerns that should still be addressed.

We would like to thank the reviewer again for her/his thorough analysis of our manuscript and many useful comments, which let us to substantially improve the manuscript.

Remaining concerns:

1. Previous concern 1: It may just be the way the relevant sentence is written, but the authors appear to claim in the text that acivicin "slightly" decreased the number of progeny. However, extended figure 10a shows that acivicin decreases brood size to about 28%, and that the difference is significant. They should insert the word "significantly" after "progeny" in the relevant sentence. In any event, their strongest argument that their lifespan extension does not involve reproductive suppression is its independence from *skn-1*, which does suffice.

We modified the text as advised.

2. Previous concern 2: the authors in their response claim that 40 μ M FuDR does not affect lifespan and does not reverse the NAC effect. Could the authors provide or point out the data supporting this claim? This is important because they used 40 μ M FuDR to measure the effect of NAC on the transcriptome.

Several previous publications studied the effect of FUDR on *C. elegans* aging. It has been shown that while low doses had insignificant effect on wt worms (PMID: 21893079), the higher dose (400 μ M) modestly extended the lifespan (PMID: 153363). In our hands, 100 μ M FUDR extended the lifespan (compare untreated controls in Fig. 1a and Extended Data Fig. 1b). The comparison between the lifespans of untreated controls in Figs 1c (23.04 ± 1.18 days) and 1e (23.38 ± 1.77) suggests that 40 μ M FUDR does not affect aging (see Supplementary Table 5).

3. Previous concern 10: Could the authors clarify how they calculated cell survival in figures 7f and 7g? Did they use the number of cells after acivicin treatment but before stress, or the number of cells they plated? 24 hrs is long enough for most cells to go through a cell cycle, and cell numbers double (extended figure 11a). Besides, 24hr acivicin treatment seems to inhibit cell proliferation and alters the number of cells (extended figure 11a). Why did the authors use such a long treatment? Is it because shorter treatments do not affect GSH uptake? Is it established that the condition they used decreases intracellular GSH or reduced thiol levels? Did they also use the same condition for RNA-Seq? Survival is misspelled in panel f.

We calculated cell survival in Figs. 7f and 7g as a ratio between the number of stressed and unstressed live cells. For unstressed cells, we used a number of cells after acivicin treatment, but before stress. We think this is an appropriate way to calculate survival, as acivicin inhibits cell proliferation but does not kill those cells (Extended Data Fig 11). To be fully transparent, we included the cell count results in **Extended Data Figure 11c and d**.

According to the published data (PMID: 15469854), it appears that the longer treatments with acivicin are more effective at inhibiting GSH uptake. The acivicin concentrations and incubation period were selected based on previously published data, which established the decrease of cellular thiols under these conditions (PMID: 15469854, PMID: 9178957, PMID: 15469854, PMID: 2870063). The time point of 24 hr was chosen to obtain the strongest acivicin effect at the concentrations used. Same conditions were used for RNA-Seq and stresses.

4. It has been shown that moderate SKN-1 overexpression increases lifespan (Tullet et al, Cell, 2008), and the Choe lab reported previously that *wdr-23* mutation or knockdown increased lifespan and GSH levels in a *skn-1*-dependent manner (PMID 26056713). Do the authors have thoughts on why *wdr-23* KD shortened lifespan here (supplementary table 5)? The authors indicate that NAC suppresses *skn-1* to decrease lifespan. Why did NAC not shorten the lifespan of *wdr-23* KD worms? Also, why was this lifespan experiment performed at 25C instead of their typical temperature of 20C?

The original screen, which discovered the *wdr-23* RNAi phenotype, was performed in the *eri-1(mg366)* strain, to improve RNAi for all cell types including neurons (PMID: 17411345). As *eri-1* is a ts mutation, we did the lifespan experiment at 25°C starting from L4. We have verified the correct sequence of *wdr-23* RNAi plasmid and detected a very strong upregulation of *gst-4::GFP* expression upon *wdr-23* knockdown. Thus, we know that RNAi was efficient and SKN-

1 activated. However, in our hands *wdr-23* RNAi did not increase the lifespan. We consider two non-mutually exclusive explanations:

First, it is possible that we did not reproduce the exact conditions used by other laboratories. For example, both the Ruvkun (PMID: 17411345) and Choe (PMID: 26056713) labs used 400 μ M of FUDR in their lifespan assays. We did not use FUDR in this experiment as we knew it interferes with the NAC effect (Extended data fig. 1b).

Second, it is possible that RNAi in our hands worked too well and SKN-1 was activated more strongly. As noted by the reviewer, only a moderate overexpression of *skn-1* extends the lifespan, while a strong overexpression can be toxic (PMID: 18358814). Moreover, a constitutive activation of SKN-1 shortens the lifespan (PMID: 23040073, PMID: 24440036). *wdr-23(tm1817)* strain is not a null allele. Staab et al (PMID: 23555279) demonstrated that *wdr-23(tm1817)* expresses a truncated transcript, in which exons six and eight were spliced together. Thus, *wdr-23(tm1817)* can be a partial loss of function mutant and activates SKN-1 only mildly. Together these results argue that a moderate activation of SKN-1 is beneficial for the lifespan, whereas stronger activation can be detrimental. Our results fit this hypothesis. Whereas NAC shortens the lifespan of worms treated with an empty vector control RNAi (Extended Data Fig. 5c), it slightly increases the lifespan of worms on *wdr-23* RNAi (Extended Data Fig. 5d). This suggests that SKN-1 was activated too strongly on *wdr-23* RNAi and that NAC treatment inhibited its activity thereby extending the lifespan.

This interesting problem deserves further investigation. However, because it is not the major point of our manuscript, we do not feel it justifies a separate discussion, unless the reviewer thinks otherwise.

5. In the last sentence of page 11, the statement that “the downregulation of *daf-16*- and *skn-1*-mediated response shortens the lifespan of animals fed LB by the same mechanism as does NAC” is not very convincing. First, NAC still shortens the lifespan of *daf-16* mutant worms (figure 3). Second, a large number of genes changed by NAC in the transcriptome are not *DAF-16* or *SKN-1* targets (figure 3c). I agree that *skn-1* suppression may mediate the effect of NAC to some extent, but their statement is too strong.

Thank you for this comment. In the revised manuscript we changed the sentence to indicate that *SKN-1* inhibition may not be the only reason for LB-mediated life shortening.

6. The authors establish that high levels of antioxidant thiols reduce lifespan by inhibiting *SKN-1* (*NRF2*), but the conventional thinking would be that *SKN-1* otherwise increases lifespan by bolstering antioxidant defenses. This suggests that other functions of *SKN-1* might be the most pertinent for aging/longevity. It would be good for the authors to point this out more explicitly and if they like speculate on what these functions might be.

We appreciate this suggestion. We believe that the major benefit of *SKN-1* is not that it increased GSH production, but the expression of detoxification genes like GSTs that help to get rid of xenobiotics and toxic metabolites, whereas GSH can interfere with healthy ROS signaling. We have added this to the discussion.

Minor issues:

1. Previous concern 5: In the response, the authors state that 15 mM NAC worms show no difference in *tbb-6::GFP* intensity compared to untreated worms. This is supported by the image

in figure 6b and the quantification graph. However, in the manuscript (on page 14) the authors claim that NAC decreased *tbb-6::GFP* fluorescence. They should clarify if they intend this to apply only to acivicin-treatment conditions.

Indeed, NAC decreases the *tbb-6* expression only in acivicin-treated worms. We modified the sentence accordingly.

2. Can the authors clarify the statement “acivicin slightly delayed development” in more detail? This is relevant to the increased stress resistance of acivicin-treated worms than untreated worms. This is also relevant to the lifespan assay because acivicin treatment increased lifespan about 2 days when started at eggs than L4 (Extended table 5).

We always count the days of the lifespan starting from L4 (day 0). When we treated worms with acivicin from eggs we started the count from the day we picked L4 worms from unsynchronized plates. Thus, the developmental delay is not included in the acivicin-mediated lifespan increase. Similarly, we picked L4 worms and let them grow till A2 before stressing them with heat or tunicamycin. In the revised manuscript we added a graph (new Extended Data Fig. 10a) demonstrating the acivicin induced developmental delay.

3. In the abstract, the statement that “the long-lasting effect of GSH or NAC on healthy animals has not been investigated” is not accurate. It has been reported that NAC has no effect on the lifespan of wild-type worms in the presence of FuDR (Wei and Kenyon, PNAS, 2016). Also, as the authors mentioned, NAC abolishes the beneficial effects of exercises (their reference 4). I recommend “the long-lasting effect of GSH or NAC on healthy animals has not been intensively investigated”.

We changed the sentence in the abstract as requested by the reviewer.

4. The sentence in the second and third lines of page 6 that “Thiol level in worms reared on LB+5 mM diamide was very similar to that in worms reared on DB (Fig 2d and e), indicating that the further depletion of thiols is not beneficial” needs to be explained in more detail.

In the revised manuscript we modified this sentence to clarify it.

5. For lifespan results in figure 5b-e, extended figure 5c-d, are the red numbers in the graphs changes in % or absolute numbers of days?

Thank you for pointing out our mistake. The numbers are changes in %. We corrected this in the revised manuscript.

6. The statement on page 15 that “Accordingly, feeding worms with DB - a diet deficient in low molecular weight (LMW) thiols, markedly increases the lifespan of wt, but not *nuo-6* worms (Extended Data Fig. 8a and b)” is not clear. According to the extended data Fig. 8, DB still increased the lifespan of *nuo-6* worms, albeit to a lesser extent.

In the revised manuscript we modified the sentence as suggested by the reviewer.

7. In the first line of page 18, the authors need to clarify what is GGT1.

In the revised manuscript we spell out GGT1 is Gamma-glutamyltransferase 1.

8. In addition, why did the authors use one-tailed t-test in their lifespan statistical analysis? One-tailed t-test hypothesizes that the change will only go to one direction, while two-tailed t-test is more proper here because the investigators would not know in which direction the change might go (whether lifespan will increase or decrease).

We recalculated the p -values as requested by the reviewer.

Reviewer comments, fourth round –

Reviewer #1 (Remarks to the Author):

The authors have appropriately revised their manuscript.

Reviewer #2 (Remarks to the Author):

I am satisfied with the response to my second set of comments, and very pleased that the authors worked hard to address all reviewer suggestions.